# A Fast Method of High-Frequency Induction Cladding Copper Alloy on Inner-Wall of Cylinder Based on Simulation and Experimental Study

**Longlong He** *, **Yafei Wang, Ruiyu Pan, Tianze Xu, Jiani Gao, Zhouzhou Zhang, Jinghui Chu, Yue Wu** * **and Xuhui Zhang**

Shaanxi Key Laboratory of Mine Electromechanical Equipment Intelligent Detection and Control, Xi'an University of Science and Technology, Xi'an 710049, China; wyf20160423@163.com (Y.W.); pry17835972904@163.com (R.P.); x13835872797@163.com (T.X.); gjn20190326@163.com (J.G.); helonger@gmail.com (Z.Z.); chujh@xust.edu.cn (J.C.); zhangxh@xust.edu.cn (X.Z.)
* Correspondence: hell@xust.edu.cn (L.H.); wuyue1992@xust.edu.cn (Y.W.)

**Abstract:** To quickly repair the inner-surface damage to the hydraulic support cylinder caused by frequent scratches, corrosion, and wear in the process of fully mechanized coal mining, this paper proposes a method of high-frequency induction cladding (HIC) copper alloy on inner-wall of cylinder (IWC) to improve the corrosion, sealing and pressure retention performance of hydraulic cylinders combined with numerical simulation and experimental study. Firstly, a numerical temperature field model for HIC of copper alloy on the IWC is established to investigate various distribution patterns and influencing factors including frequency of induction heating, gap between coil and cladding, power supply rating, cladding thickness and side length of square section of induction coil, etc. Subsequently, an HIC test experiment is conducted to rigorously validate the numerical temperature field model and the experiment employs a meticulously collected dataset of temperature measurements, confirming the model's accuracy and consistent alignment with anticipated changing trends. In addition, the experiment results were verified through microstructure observation, microhardness testing, friction-wear testing, and electrochemical corrosion parameters, which shows that the factors of induction heating frequency and others have obvious effects on the temperature field distribution of HIC copper alloy on the IWC. Under these working conditions (cladding thickness 1.5 mm, power supply rating 120 kW, heating frequency 120 kHz, gap between the cylindrical workpiece and the induction coil 3 mm, induction coil cross-sectional side length of 10 mm), the thermal impact on the cylinder barrel matrix is minimal, the metallurgical bonding between the cladding layer and the matrix is good, and there are no over burning and porosity defects.

**Keywords:** inner wall of cylinder; high frequency induction cladding; metallurgical combination; performance analysis





## 1. Introduction

Hydraulic support is one key piece of equipment in modern coal mining and it plays a crucial role in providing a safe working environment for underground operations [1,2]. Due to poor working conditions, the inner wall of the hydraulic support cylinder often suffers from varying degrees of corrosion, erosion and other damages [3,4], as shown in Figure 1. In order to enhance the wear resistance and corrosion resistance of hydraulic cylinder barrels, and to improve their operational efficiency and cost-effectiveness, numerous scholars from both domestic and international backgrounds have dedicated their efforts to researching the metallurgical bonding technologies between cylinder barrel substrates and claddings [5–7]. Currently, although there have been relatively numerous studies focused on optimizing cladding structure and processes to enhance substrate performance, research on the induction melting cladding process of copper alloys on the inner walls of cylinders

remains relatively limited [8]. Common metallurgical bonding technologies between barrel substrates and claddings include overlay welding [9,10], thermal spraying [11,12], electric spark deposition [13,14], and laser cladding [15–17].

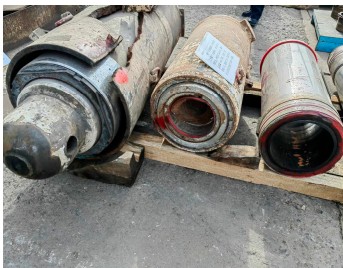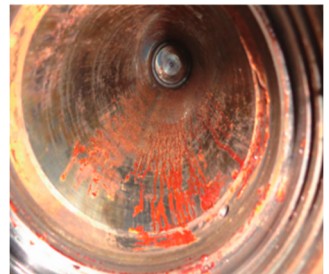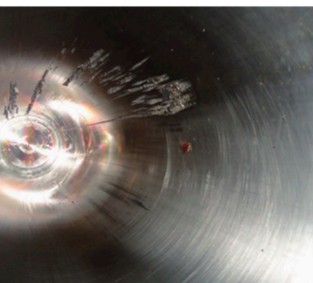

**Figure 1.** Damage to the inner wall of the hydraulic support cylinder.

Shi et al. utilized plasma transferred arc technology to deposit cobalt-based claddings with three different tungsten carbide content on the surface of low-temperature steel E32. The study revealed that by conducting PTA welding on the surface of E32 steel, a surface-modified cladding consisting of tungsten carbide reinforced cobalt was obtained. The cladding exhibited a dense structure, high hardness, and excellent low-temperature wear resistance [18]. Zou et al. conducted a simulation study using Fluent to investigate the deposition and solidification processes of droplets colliding with the substrate in the arc spraying process. By optimizing the parameters, aluminum, zinc-aluminum alloy, and zinc-aluminum alloy pseudo-alloy claddings were prepared using the arc spraying method, and the performance of the claddings was analyzed. The results indicated that the arc-sprayed Zn-Al pseudo-alloy cladding exhibited excellent wear resistance and corrosion resistance [19]. Li et al. successfully prepared W-Mo high melting point composite claddings on the surface of PCrNi3MoVA steel using the ZY-10 spark deposition/cladding equipment with ultra-pure argon gas as the shielding gas. The study found that the thickness of the W-Mo high melting point composite cladding increased with the number of depositions and deposition voltage. The composite cladding and substrate were melted together, exhibiting good metallurgical bonding characteristics. Increasing the thickness of the composite cladding effectively suppressed the increase in the erosion rate of the coated specimens [20]. Zhu et al. proposed the use of liquid chromatography-assisted in-situ induction heating laser cladding (LC-IH) technique to prepare high-performance 18Ni300 claddings on the surface of bainitic rail wings. The results showed that after depositing 18Ni300 claddings using the LC-IH method, the flexural strength and impact toughness of the bainitic rail wings increased by 4.2% and 20.0%, respectively. Additionally, the wear resistance of the LC-IHed rail was improved by 3.3 times compared to the untreated rail and 1.3 times compared to the LCed rail. Furthermore, the plastic deformation and fatigue crack propagation depth were significantly reduced in the LC-IHed rail [21].

While the techniques mentioned above undoubtedly have the potential to enhance substrate performance, it is crucial to recognize and address their inherent limitations when implementing them in practical applications. Weld overlay can achieve localized repairs, but the bonding strength between the overlay and the substrate is relatively low, which may lead to the generation of cracks and bubbles, thereby affecting the repair effectiveness [22]. Thermal spraying operation is relatively simple, but the bonding strength between the cladding and the substrate is low, and the porosity rate is high. Additionally, a large amount of dust and fumes are generated during the spraying process, causing environmental pollution [23]. Although Electrical Discharge Machining (EDM) deposition technology enables the fabrication of high-thickness and continuous claddings, the rapid heating and rapid cooling characteristics associated with this process may result in the occurrence of numerous cracks and severe element segregation within the claddings [24,25]. Although laser cladding technology can achieve high deposition rates, it is hampered by expensive equipment, challenging operations, low production efficiency, and higher

cladding costs [26–28]. In comparison, HIC offers advantages such as precise control, environmental friendliness, high heating efficiency, lower operational complexity, lower repair costs, and strong bonding between the cladding and the substrate [29–32].

Induction cladding technology is an efficient method for preparing claddings that enhance the surface performance of materials [33]. By utilizing an induction coil, alternating current is employed to generate an alternating magnetic field in close proximity to the metal's surface, thereby inducing eddy currents. These eddy currents, in turn, produce heat that leads to the fusion of the conductive cladding material. Prior to the induction process, the self-fusing metal powder cladding can be applied onto the heated substrate surface using cold spray or thermal spray techniques. Consequently, a sequence of physicochemical reactions takes place between the molten cladding material and the substrate, resulting in the formation of a strong metallurgical bond between the cladding and the substrate [34]. This surface cladding technique enables the production of high-performance claddings. For example, Dong et al. utilized high-frequency induction cladding technology to prepare NiTiFe alloy claddings on the surface of HT300 substrate in order to enhance the wear resistance of the metal material. Subsequently, the claddings were subjected to metallographic analysis and friction-wear tests. The research findings revealed that the cladding exhibited a dense microstructure without cracks or voids, indicating excellent formation quality. Furthermore, the cladding demonstrated outstanding resistance to frictional wear [35]. In order to improve the surface performance of 304 stainless steel blade tips, Bing et al. employed the ultra-high-frequency induction heating technology to coat the substrate with 316L stainless steel wire. The study investigated the microstructure, tribological properties, and corrosion behavior of the claddings, and identified the optimal combination of process parameters. The research findings demonstrated that the 304/316L cladding significantly enhanced wear resistance and corrosion resistance [36]. Xiong et al. utilized high-frequency induction cladding technology to prepare TiC/Ni composite claddings on the surface of sucker rods. The results showed that the post-solution aging of the claddings did not exhibit significant defects. The claddings mainly consisted of γ-Ni solid solution, Ni compounds, CrB, carbides, and δ phase hard particles. After aging, the maximum hardness of the cladding reached 928.2 HV0.2, which was 4% higher than before aging. The minimum wear amount was 48.9 mg, reducing 62% compared to the substrate. After 4 h of aging, the claddings exhibited the best overall performance [37].

Wang conducted an analysis of the surface corrosion mechanism of hydraulic support column materials and conducted research on three widely used techniques: copper melting, electroplating, and laser cladding. She summarized the comprehensive performance requirements for underground environmental protective claddings and proposed the potential for developing surface thin film protection technologies [38]. To address the issues of insufficient bonding strength and corrosion resistance of copper plating layers on the inner wall of hydraulic cylinders, Jiao et al. selected quenched and tempered 27SiMn steel as the substrate and used arc cladding technology to create copper melting layers on the inner wall of hydraulic cylinders. The research results showed that the copper melting layers with certain alloying elements exhibited excellent anti-friction and wear resistance properties under dry friction and emulsion conditions [39]. Therefore, this study proposes a method of HIC of copper alloys on the inner wall of cylinder sleeves. It conducts numerical simulations using finite element software (COMSOL 6.0) and validates the results through experiments. The study also investigates the microstructural properties of the cladding layers, clarifies the metallurgical bonding mechanisms between the induction-clad layers and the substrate, and explores the effects of different process parameters (current frequency, coil gap, power supply rating, cladding thickness and coil cross-sectional dimensions) on the performance of the cladding layers. The aim is to obtain low-cost and high-quality copper alloy cladding layers and provide a theoretical basis for achieving high-quality metallurgical bonding between the inner wall of cylinder sleeves and the claddings.

The remaining sections of this article are organized as follows: Section 2 provides a detailed description of the establishment of the numerical model and the simulation

process of the temperature field. In Section 3, HIC experiments were conducted based on the numerical simulation results, and the performance analysis of the experimental samples was carried out. Finally, Section 4 summarizes the main conclusions of the study.

## 2. Numerical Simulation of Copper Alloy Induction Cladding on the Inner Wall of Cylindrical Workpieces

### 2.1. The Principle of High-Frequency Induction Heating

The process flow of HIC technology is as follows: Firstly, high-quality alloy powder is mixed with binders and fluxes, and the mixture is cold-sprayed onto the surface of the workpiece. Next, the workpiece is heated using high-frequency induction heating. High-frequency induction heating utilizes principles such as electromagnetic effects and skin effect. The alternating current generated by the heating power supply produces an alternating magnetic field in the induction coil, which interacts with the heated workpiece. This interaction induces eddy currents inside the heated workpiece, flowing in the opposite direction to the high-frequency alternating current in the coil, known as the eddy current effect. Under the thermal effect of the eddy current, the cladding layer rapidly heats up, and the alloy powder quickly and completely melts, thereby forming a metallurgical bond with the substrate [40,41]. Figure 2a illustrates the principle of electromagnetic induction heating.

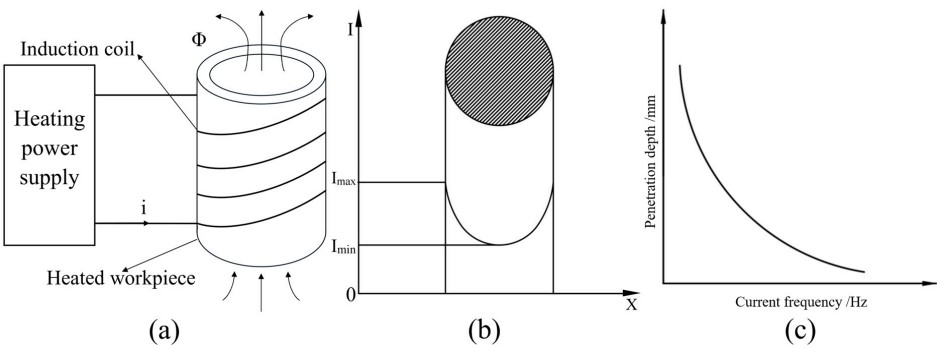

**Figure 2.** (**a**) Schematic diagram of electromagnetic induction heating. (**b**) Schematic diagram of skin effect. (**c**) Schematic diagram of penetration depth.

The induction heating system mainly consists of a heating power supply, heating coil, and the workpiece being heated. In the process of induction heating, when alternating current with a certain frequency passes through the conductor, most of the current is concentrated on the surface of the conductor due to the skin effect, resulting in a power law decline in current density from the surface of the conductor to the inside [42,43]. Figure 2b illustrates the current distribution across the cross-section of the conductor. The depth at which the current density is 36.8% of the surface value is known as the penetration depth of the current. The penetration depth of the current is calculated using Equation (1) [44].

$$\delta = 5030 \sqrt{\frac{\rho}{f \cdot \mu}} \tag{1}$$

The formula is as follows: $\delta$ represents the penetration depth in centimeters (cm); $\rho$ represents the resistivity in ohm-centimeters ($\Omega \cdot$cm); $\mu$ represents the relative permeability, which is equal to 1 for air and non-magnetic materials; $f$ represents the frequency of the power supply in hertz (Hz).

According to Equation (1), the penetration depth of the current varies with changes in current frequency, resistivity, and permeability while keeping the workpiece material constant [45]. Figure 2c illustrates the relationship among these factors. In this study, the inner surface of a cylindrical 27SiMn steel substrate was chosen as the target for cladding with a copper alloy. Physical parameters, such as thermal conductivity, specific heat capacity, relative permeability and resistivity, are obtained from the Online Materials Information

Resource—MatWeb (https://matweb.com/index.aspx, accessed on 11 March 2024) webpage. The trends of the physical parameters of the cylindrical steel substrate with changing temperatures are shown in Figure S1, while Figure S2 demonstrates the trends of the physical parameters of the copper alloy with temperature variations. Since the copper alloy is a non-magnetic material, the hysteresis heating effect can be neglected, and only the thermal effect generated by eddy currents is considered. The object being heated is the inner wall of the cylindrical workpiece, and to protect the integrity of the cylindrical workpiece substrate, it is necessary to minimize the heating time and reduce the penetration depth of the current.

## 2.2. Establishment of Numerical Model for Induction Cladding

In order to establish the numerical model of HIC, COMSOL Multiphysics software is used in the experiment, which is famous for being good at simulating and modeling various physical fields. Cooling water flows inside the induction coil, and a copper-based alloy powder is used for the cladding layer. Because the melting point of white copper alloy is between 1000 °C and 1083 °C, and the melting point of 27SiMn steel is about 1500 °C, the heating temperature in the process of induction cladding is set between 1000 °C and 1100 °C. This not only ensures the complete melting of copper alloy coating, but can also cause serious thermal damage to the substrate. Figure 3 illustrates a two-dimensional schematic of the induction cladding process, while the three-dimensional model used for numerical simulation analysis is shown in Figure 4a.

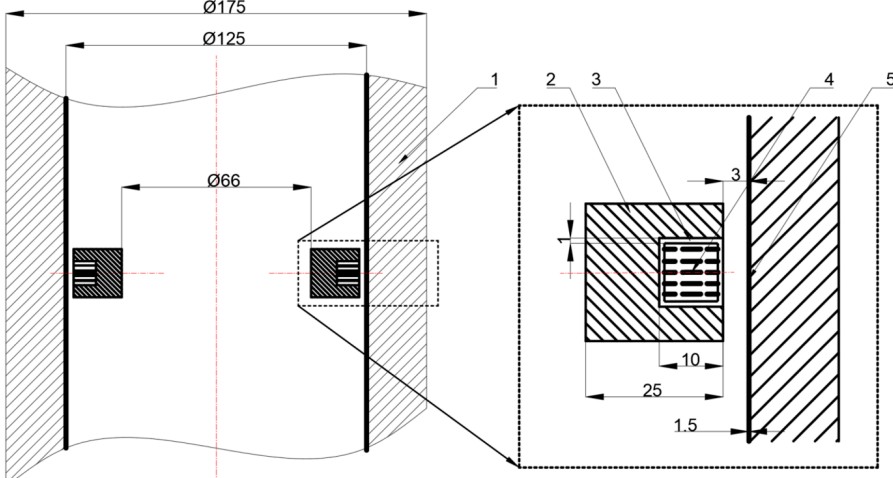

1-Cylinder body, 2-Magnetizer, 3-Induction coil, 4-Cooling water, 5-Copper alloy cladding

**Figure 3.** Induction heating schematic diagram.

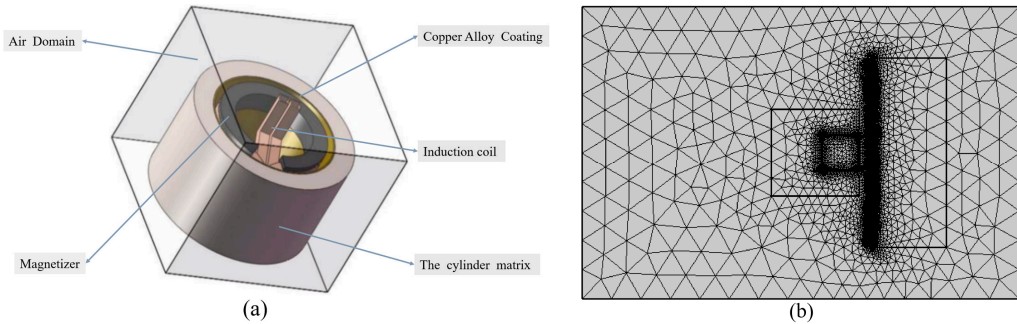

(a)                (b)

**Figure 4.** (**a**) Induction heating three-dimensional model. (**b**) Finite element model mesh division.

During the numerical modeling of the induction cladding process, the induced eddy currents generated in the copper alloy cladding layer primarily concentrate near the inner surface of the cladding layer due to the skin effect and proximity effect. To accurately

describe the distribution of induction heating, the cladding layer of the copper alloy is divided into at least four layers of volumetric elements within the depth of current penetration. The air domain elements are controlled using a physics-controlled mesh. The specific division scheme is shown in Figure 4b. In the analysis of induction cladding heating, the elements used and their corresponding material parameters are presented in Table 1.

**Table 1.** System Composition Corresponding Material Parameters.

| System Composition | Description of System Composition |
|---|---|
| Cylinder base | Material is 27SiMn steel, outside diameter is 175 mm, inside diameter is 125 mm |
| Cladding layer | Material is a white copper alloy, with a thickness of 1.5 mm |
| Induction coil | Material is a T3 square copper tube, side length is 10 mm, wall thickness is 1 mm, cooling water flow rate is 2 kg/min |
| Magnetizer | Material is a square silicon steel sheet, side length is 25 mm, near the copper alloy layer with an edge length of 11 mm square notch |
| Air domain | Set material to air, relative magnetic permeability to 1, electrical conductivity to 0, other parameters are given by the material library |

*2.3. Analysis of Temperature Field Results on the Inner Wall of Cylindrical Workpieces under Corresponding Process Parameters*

In order to adjust the experimental parameters and refine the numerical simulation model, actual parameters of HIC of copper alloy on the inner surface of a cylindrical workpiece were used, and the experimental results were compared with the numerical simulation results. The process parameters are as follows: power supply power of 120 kW, current frequency of 120 kHz, HIC time of 10 s, coil outer diameter of 116 mm, square cross-section of the coil with a side length of 10 mm, one turn of the coil and cladding thickness of 1.5 mm. The thermal radiative coefficients for the cylindrical steel substrate and the copper alloy cladding are 0.7 and 0.8, respectively. The numerical simulation was performed using natural convection conditions with a time step of 0.1 s for a total of 100 load steps. Based on this, an analysis of the transient temperature field during the HIC process was conducted.

Using the post-processing module of COMSOL Multiphysics, the temperature distribution of the cylindrical workpiece at different times was obtained, as shown in Figure 5. From Figure 5a–f, it can be observed that the copper alloy cladding is initially heated, and then heat is transferred to the cylindrical steel substrate. During the induction heating process, the irregular magnetic field lines generated by the induction coil result in an irregular distribution of high-temperature regions in the cladding. The thermal conductivity of the copper alloy is higher than that the 27SiMn steel substrate, leading to a greater temperature gradient in the substrate compared to the cladding. When the heating reaches 9 s, the copper alloy cladding begins to melt, and by 9.5 s, the copper alloy is completely melted. During the initial 9 s of heating, the maximum temperature of the cladding rises by nearly 900 °C, but within the 0.5 s phase of copper alloy melting, the maximum temperature only increases by 30 °C. This phenomenon of reduced temperature rise is caused by the increased electrical resistivity, increased specific heat capacity, and latent heat effects associated with phase change.

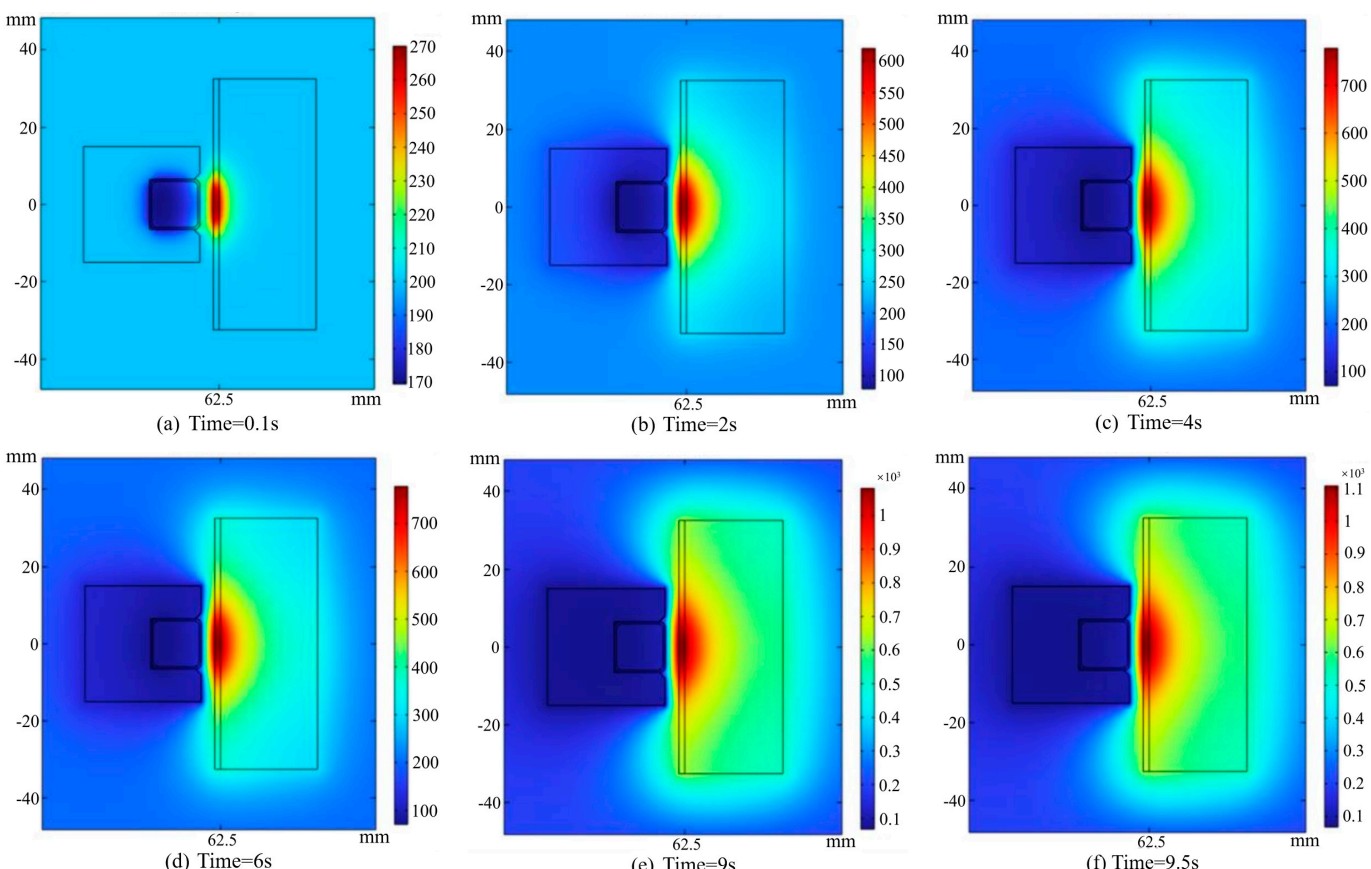

**Figure 5.** The temperature distribution of a cylindrical workpiece at different times.

In order to study the temperature variation at specific locations on the cylindrical workpiece and observe the HIC cycle, nodes were selected at the positions shown in Figure 6. These nodes are located on the surface of the copper alloy cladding (points A, B, and C), the interface between the substrate and the cladding (points D, E, and F), and the outer surface of the substrate (points H, I, and J). Figure 7 displays the temperature curves of these nodes as they vary over time during the HIC process. It can be observed from the figure that the temperature curves of points A and E are close and exhibit similar trends, while the curves of points B, C, D, F, I and J overlap to a large extent. At the beginning of the induction cladding process, except for points H, I, and J, the temperature of the other points rises rapidly. This is due to the skin effect and proximity effect, which result in the generation of maximum eddy current heat on the surface of the cladding, with insufficient time for heat to conduct into the substrate. After 2.5 s of heating, the rate of temperature increase slows down. This is mainly because the heat radiation and convection from the substrate to the surroundings increases, and a significant amount of heat is transferred to the substrate. When the heating and cooling processes of the cladding layer reach a dynamic equilibrium, the rate of temperature rise in the cladding slows down. Points H, I, and J are farthest from the induction coil and in direct contact with the air on the outer surface of the cylinder. Therefore, their temperatures are significantly lower compared to the other points. From Figure 8, it can be observed that the temperature distribution on the surface of the cladding is uniform, with a small temperature gradient. This is due to the relatively thin cladding thickness of 1.5 mm, the excellent thermal conductivity of the copper alloy, and the shallow penetration depth of eddy currents.

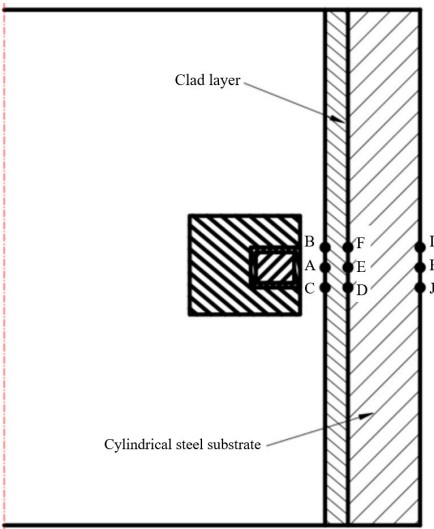

**Figure 6.** Schematic diagram of selecting nodes.

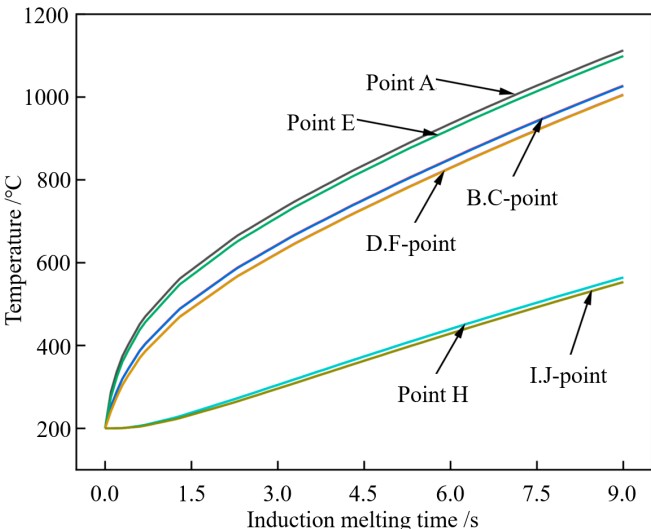

**Figure 7.** Thermal cycle curves of cylindrical workpieces at different positions.

To investigate the radial temperature distribution of the substrate and cladding at different time intervals, points A, E and H from Figure 6 were selected as observation positions. Figure 8 illustrates the radial temperature distribution at various time points. Figure 8a,b depict the temperature distribution from the outer surface to the inner surface of the cladding after 2.5 s and 9 s of heating, respectively. As the heating time progresses, the radial temperature variation within the cladding remains minimal, and there is a tendency for the temperature gradient to decrease in the later stages of induction heating. Figure 8c,d display the radial temperature distribution of the substrate after 2.5 s and 9 s of heating, revealing a much larger temperature gradient within the substrate. During the initial stage of heating, the temperature difference reaches 380 °C, and it maintains a temperature difference of approximately 502.5 °C as the process continues. The temperature of the substrate exponentially decreases from the inner surface to the outer surface during the heating process. This phenomenon is attributed to the relatively short melting time of the cladding in the induction cladding process, which effectively protects the substrate from thermal damage.

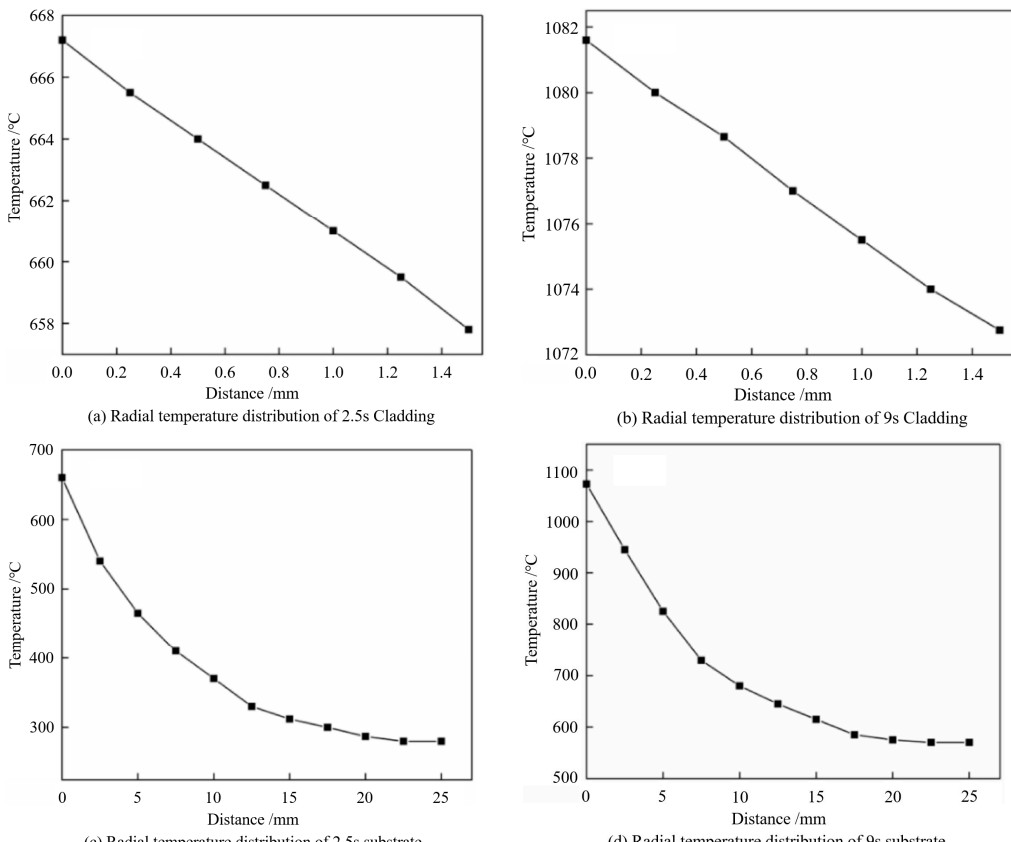

**Figure 8.** Induction melting radial temperature at different positions.

*2.4. The Influence of Different Process Parameters on the Temperature Field of Cylindrical Workpieces*

(1)    The impact of heating frequency on the temperature field

Due to the inverse relationship between the current f and the penetration depth δ, a higher current frequency results in a smaller penetration depth, leading to a thinner heating thickness of the workpiece [46]. Therefore, different current frequencies are selected under the chosen operating conditions to investigate the influence of heating frequency on high-frequency induction melting. The relevant experimental parameters and temperature field distributions at 3 s and 9.5 s of induction melting are shown in Table 2, Figures S3 and S4. Furthermore, as the current frequency increases, the non-uniformity of the substrate temperature also increases.

**Table 2.** Experimental parameters and temperature field analysis.

| Experimental Parameters | Power Rating | 120 kw | Cladding Thickness | 1.5 mm | Coil Turns | 1 |
|---|---|---|---|---|---|---|
| Temperature variation of the cylindrical workpiece during 3 s induction melting | | | | | | |
| Current frequency | 60 kHz | 80 kHz | 100 kHz | 120 kHz | 150 kHz | 200 kHz |
| Maximum temperature | 612 °C | 666 °C | 703 °C | 731 °C | 754 °C | 785 °C |
| Temperature variation of the cylindrical workpiece during 9.5 s induction melting | | | | | | |
| Current frequency | 60 kHz | 80 kHz | 100 kHz | 120 kHz | 150 kHz | 200 kHz |
| Maximum temperature | 946 °C | 1040 °C | 1100 °C | 1150 °C | 1200 °C | 1240 °C |

To understand the influence of current frequency on the temperature distribution at the metallurgical bond interface, the post-processing feature of COMSOL Multiphysics was utilized to examine the temperature distribution curve at the fully melted cladding interface, as shown in Figure 9. With current frequencies of 120 kHz, 150 kHz and 200 kHz, and cladding times of 9 s, 8 s and 7.5 s, the copper alloy claddings were completely melted.

From the graph, it can be observed that the high-frequency induction coil generates the highest temperature at the corresponding midpoint of the interface. As the longitudinal temperature gradient of the substrate increases, there is no significant variation in the longitudinal temperature at different points of the metallurgical bond interface. This is attributed to the superior thermal conductivity of the copper alloy compared to the 27SiMn steel substrate.

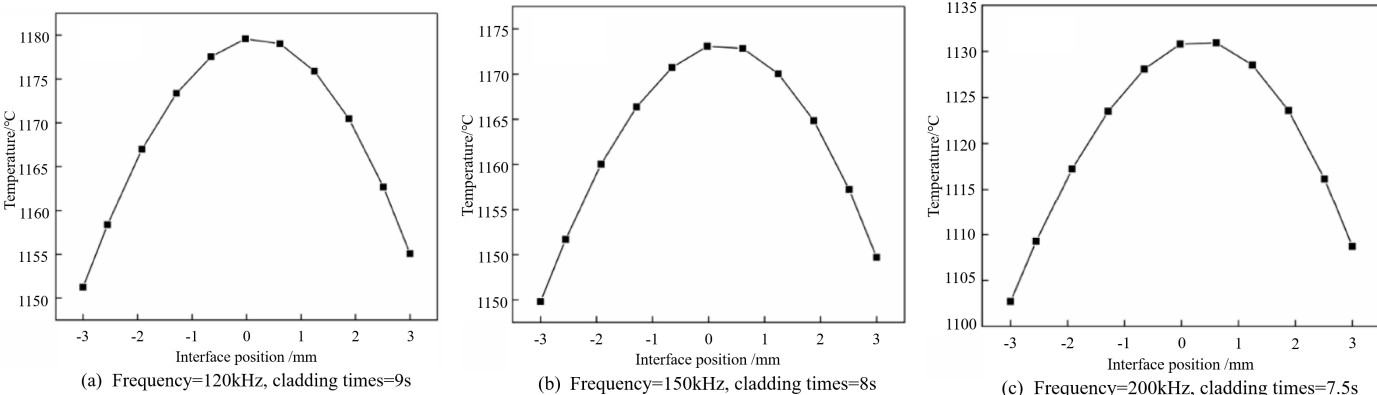

(a) Frequency=120kHz, cladding times=9s  (b) Frequency=150kHz, cladding times=8s  (c) Frequency=200kHz, cladding times=7.5s

**Figure 9.** Joint interface temperature when copper alloy cladding is completely melted.

To study the temperature variation of a cylindrical workpiece, the temperature changes at point H on the outer surface of the cylindrical workpiece, as shown in Figure 6, were selected during the induction cladding process. The temperature variation at point H during the cladding process is depicted in Figure 10. From the graph, it can be observed that at a frequency of 200 kHz and a heating time of 7.5 s, the temperature at the midpoint E of the interface reaches as high as 1100 °C, while the temperature on the outer surface of the substrate is only around 480 °C. Depending on the different current frequencies, the degree of temperature change on the outer surface of the substrate differs from the degree of change at the midpoint E of the interface. This is because the induction cladding time is relatively short, and the heat from the substrate has not fully transferred to the cladding.

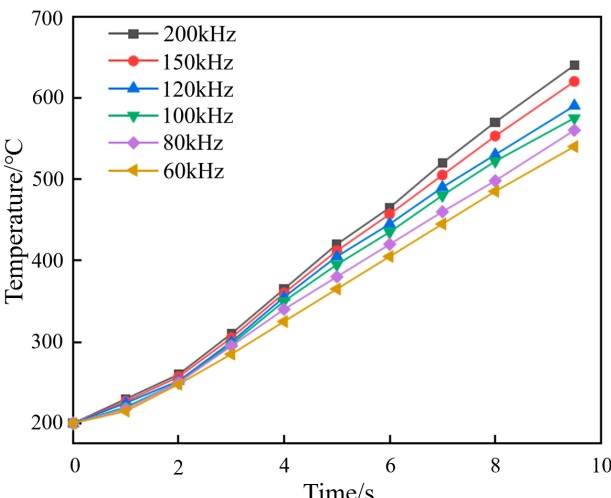

**Figure 10.** H-point temperature variation at different frequencies.

Figure 11 illustrates the temperature variation at the midpoint E of the bond interface under different frequencies. From the graph, it can be observed that at current frequencies of 120 kHz, 150 kHz and 200 kHz, the temperature at the midpoint E of the interface reaches 1100 °C within 9.5 s of induction heating, indicating complete melting of the copper alloy cladding. However, at current frequencies of 60 kHz, 80 kHz and 100 kHz,

when the induction heating time is extended to 10 s, the copper alloy cladding remains solid, indicating a failed induction cladding process. It is worth noting that due to the magnetic shielding effect of the copper alloy, reducing the heating frequency enhances the heat conduction of the cladding layer to the substrate, prolonging the melting time of the cladding. However, slow heating and prolonged melting time can potentially cause thermal damage to the substrate.

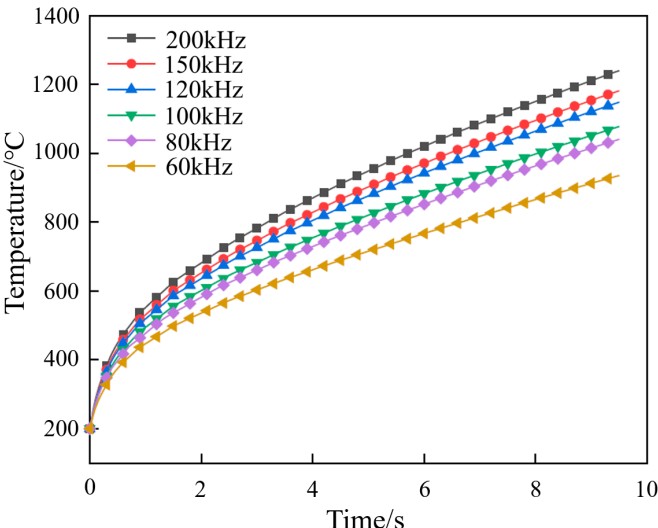

**Figure 11.** Temperature changes at the midpoint of the bonding interface at different frequencies.

(2)  The Influence of a Gap between Cylindrical Workpiece and Coil on the Temperature Field

To investigate the influence of the gap between the induction coil and the cladding on the high-frequency induction heating effect, a simulation was conducted with the following parameters: power supply power of 120 kW, current frequency of 120 kHz, induction cladding heating time of 9.5 s, and a copper alloy cladding thickness of 1.5 mm. The gap was varied by changing the outer diameter of the coil, and the specific simulation configurations are presented in Table 3.

**Table 3.** Gap simulation scheme.

| Simulated Serial Number | 1 | 2 | 3 | 4 | 5 | 6 |
|---|---|---|---|---|---|---|
| Outer diameter of coil d/mm | 110 | 112 | 114 | 116 | 118 | 120 |
| Gap/mm | 6 | 5 | 4 | 3 | 2 | 1 |

Figure S5 displays the instantaneous temperature distribution of the cylindrical workpiece after 2 s of induction heating. From the Figure S5a–c, it is evident that the maximum temperature of the cladding increases with the increase in the outer diameter of the induction coil. The influence of the induction coil's outer diameter on the induction cladding process can be primarily observed in two aspects. First, as the outer diameter of the coil increases, the gap between the coil and the cladding layer decreases, thereby improving the cladding efficiency. Second, the magnetic leakage phenomenon, as shown in Figure S5d–f, interferes with the induction cladding heating. The temperature distribution of the cylindrical workpiece with an outer diameter of the coil of 110 mm and 112 mm differs from that with an outer diameter larger than 116 mm, mainly due to variations in the cladding temperature. Although the magnetic shielding effect of the magnetic conductor reduces the impact of magnetic leakage, the magnetic leakage phenomenon strengthens with increasing gap, reducing the efficiency of high-frequency induction heating. Therefore, when cladding copper alloy claddings, it is preferable to minimize the gap while ensuring that the cladding does not come into contact with the induction coil.

Figure S6 presents the contour plots of the instantaneous temperature distribution of a cylindrical workpiece during 4 s of HIC using induction coils with different outer diameters. When the coil outer diameters are 120 mm, 118 mm, 116 mm and 114 mm, the maximum temperatures of the cladding are 1160 °C, 1040 °C, 919 °C and 766 °C, respectively. It can be observed that when the coil outer diameter is 114 mm and 116 mm, the cladding temperature does not reach the melting point. This is attributed to the significant increase in magnetic leakage due to the enlarged gap, which reduces the eddy currents generated by electromagnetic induction, insufficient to melt the copper alloy cladding. From Figure S6a–c, it is evident that when the coil outer diameter is 120 mm, the cladding starts to melt before the 4 s mark. On the other hand, when using high-frequency induction coils with a diameter smaller than 120 mm and heating for 4 s, the copper alloy cladding remains solid.

Figure 12 illustrates the temperature distribution of the melted cladding when the coil outer diameter is 120 mm and the cladding time is 3.4 s and 3.6 s. At a cladding time of 3.4 s, the cladding temperature reaches 1090 °C, which is its melting temperature. When the cladding time is extended to 3.6 s, the minimum temperature at the metallurgical bond reaches 1094 °C, indicating complete melting of the cladding. Due to the copper alloy's ability to fully melt within a short period, heat transfer causes the temperature of the cylindrical substrate to approximate a steady-state distribution. Consequently, the temperature distributions of the cylindrical workpiece at 3.4 s and 3.6 s are similar.

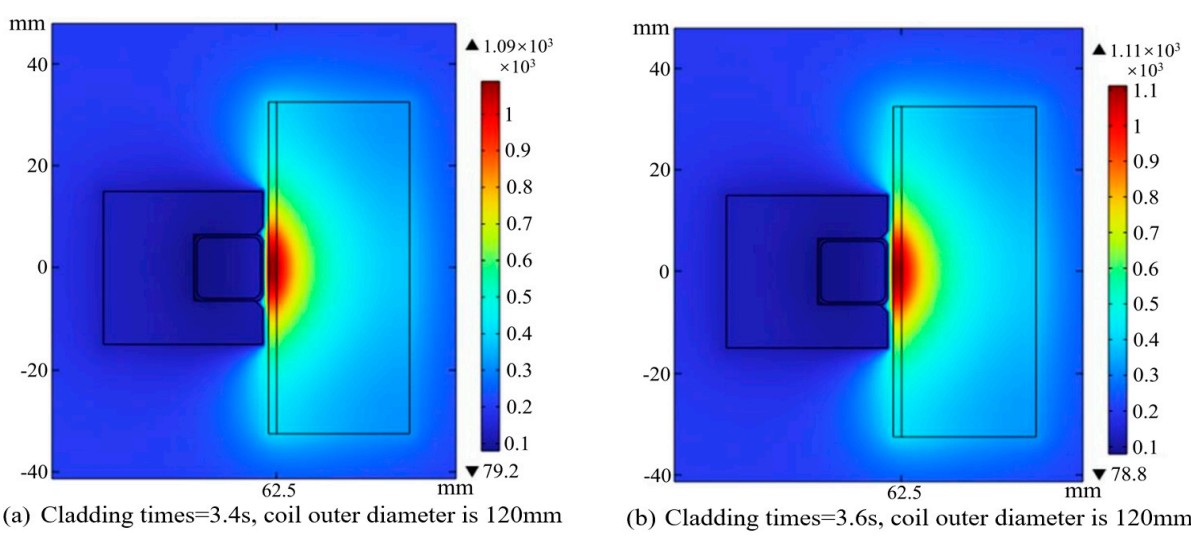

(a) Cladding times=3.4s, coil outer diameter is 120mm    (b) Cladding times=3.6s, coil outer diameter is 120mm

**Figure 12.** Melting temperature distribution of copper alloy cladding.

When the coil outer diameter is less than 110 mm, the magnetic leakage phenomenon becomes severe as the gap increases. This prolongs the time required to reach the melting temperature of the cladding and can result in thermal damage to the substrate. Figure 13 displays the temperature cycle curves at the midpoint of the interface for different outer diameters of the high-frequency induction coil. From the graph, it can be observed that when the outer diameters of the high-frequency induction coil are 120 mm, 118 mm, 116 mm and 114 mm, the cladding times required to raise the temperature to 1100 °C are 3.4–3.6 s, 4.2–4.6 s, 7.2–7.5 s and 10.0–10.5 s, respectively, satisfying the complete melting requirement of the copper alloy cladding. Additionally, when the coil outer diameter is greater than or equal to 116 mm, the copper alloy cladding can be heated to complete melting within 10 s without causing thermal damage to the cylindrical steel substrate.

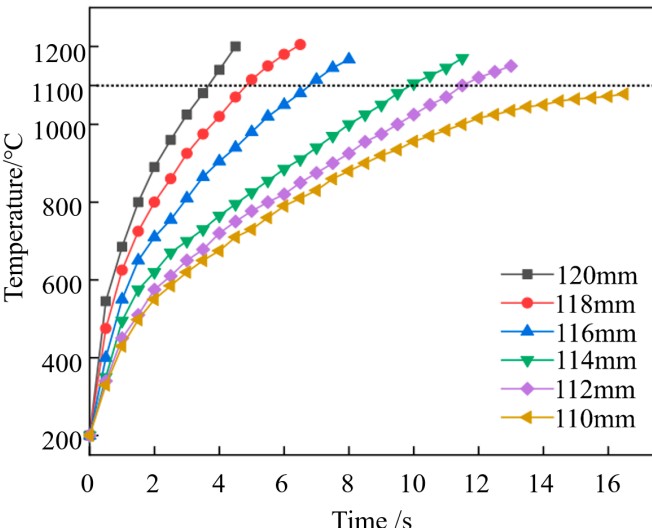

**Figure 13.** Temperature curve of point E under different induction coil outer diameters.

(3)    The impact of power supply power on the temperature field

To investigate the influence of different power levels on the effectiveness of high-frequency induction heating, numerical simulations were conducted with power levels set at 80 kW, 100 kW, 120 kW and 140 kW, while keeping the heating frequency at 120 kHz, the cladding thickness at 1.5 mm, and the gap between the induction coil and the cladding at 3 mm. When simulating a power level of 140 kW, the temperature distribution at any time and location within an 8 s cladding period for the cylindrical workpiece and the copper alloy cladding is illustrated in Figure 14. At cladding times of 3 s and 3.8 s, the maximum temperatures of the cladding are 999 °C and 1089 °C, respectively, which have not reached the melting point of the copper alloy. At a cladding time of 7.5 s, the maximum temperature at the metallurgical bond interface reaches 1400 °C, close to the melting point of the 27SiMn steel substrate. This indicates that the time required to achieve the desired cladding temperature decreases with increasing power levels. To ensure the cladding effectiveness, the subsequent cladding time should be controlled within the range of 3.8 s to 7.5 s.

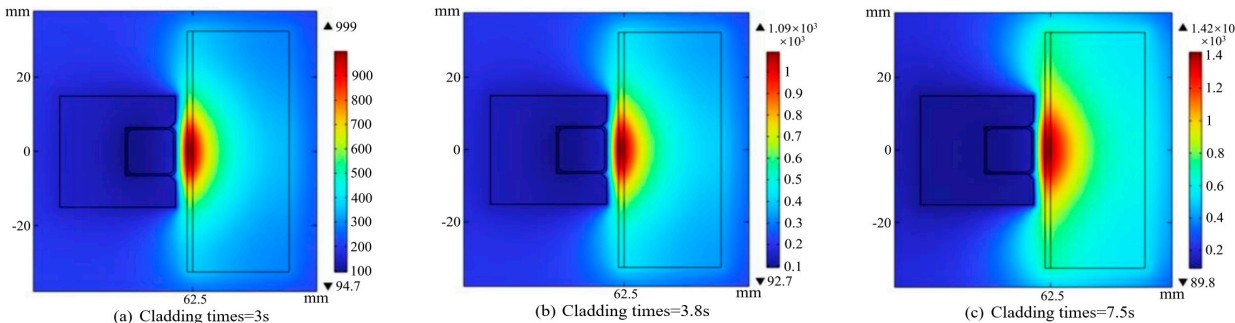

**Figure 14.** Temperature distribution at different melting times at a power of 140 kW.

In the case of a power level of 140 kW, the temperature cycle curves for various points in Figure 6 are illustrated in Figure S7. Referring to Figure 7, it can be observed that the time required for the cladding to reach the melting point decreases with increasing power levels. At a cladding time of 3.1 s, the maximum temperature of the cladding reaches 1090 °C, indicating the initiation of melting. As the cladding time extends to 4.0 s, the temperature at the bond interface reaches 1123 °C, indicating complete melting of the copper alloy cladding.

When the power level is set to 120 kW, the temperature variation of the cylindrical workpiece during a 10 s induction cladding process is shown in Figure 15. At 3 s of heating, the maximum temperature on the cladding surface reaches 865 °C, and different positions on the cylindrical workpiece exhibit varying temperature gradients. At 5.4 s of heating, the maximum temperature of the workpiece reaches 1095 °C, indicating the onset of cladding melting. At 9.8 s of heating, the maximum temperature at the metallurgical bond interface rises to 1400 °C. From the graph, it can be observed that when the power level is 120 kW, the rate of temperature increase in the heated workpiece is slower compared to the case of 140 kW.

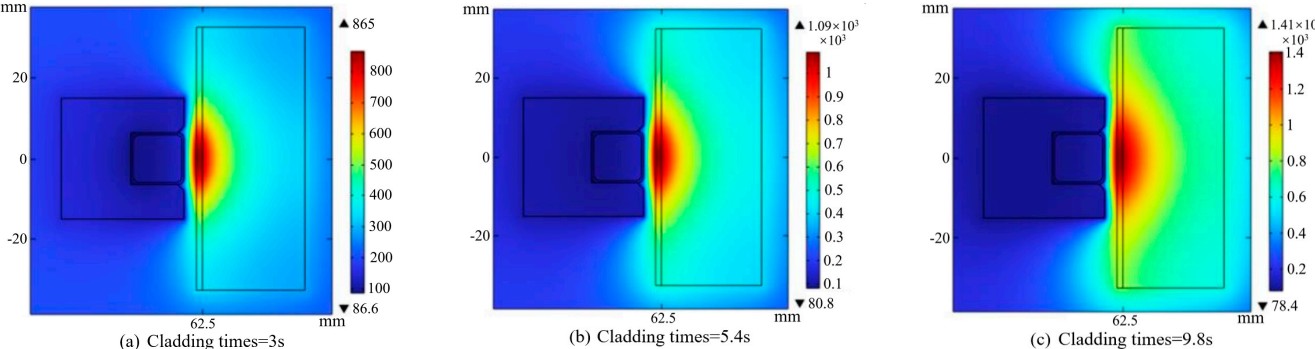

**Figure 15.** Temperature distribution at different melting times at a power of 120 kW.

The temperature cycle curves for various points on the cylindrical workpiece at a power level of 120 kW are shown in Figure S8. From the graph, it can be observed that at a cladding time of 5.4 s, the cladding begins to melt. As the cladding time extends to 9.8 s, the maximum temperature on the cylindrical workpiece reaches 1420 °C, indicating that the induction cladding heating time can be adjusted between 5.4 s and 9.8 s. With prolonged induction heating time, the temperature at point H gradually increases but with a lower rate of increase. This is because the short induction cladding heating time allows radial heat transfer to dominate on the cylindrical steel substrate.

The temperature variation of various points on the cylindrical workpiece during a 15 s induction cladding process at a power level of 100 kW is shown in Figure 16. From the graph, it can be observed that at 5 s of heating, the maximum temperature on the cylindrical workpiece reaches only 903 °C. At 7.8 s of heating, the maximum temperature on the workpiece rises to 1095 °C, indicating the onset of cladding melting. At 13.9 s of heating, the maximum temperature at the metallurgical bond interface reaches 1413 °C. This indicates that during the initial stage of heating, the heat is primarily transferred radially from the inner surface of the cladding to the substrate. The heated cylindrical steel substrate experiences both radial and axial heat transfer, with axial heat transfer dominating.

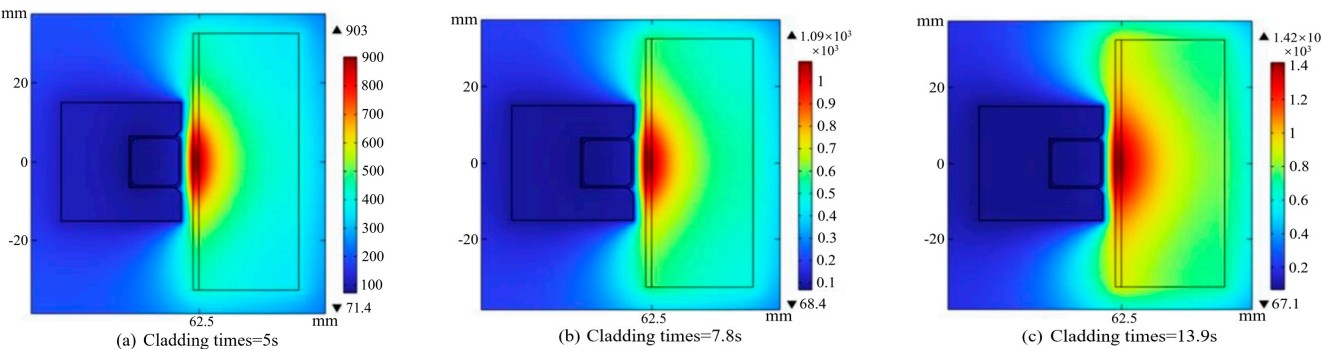

**Figure 16.** Temperature distribution at different melting times at a power of 100 kW.

The temperature cycle curves for various points on the cylindrical workpiece at a power level of 100 kW are shown in Figure S9. From the graph, it can be observed that compared to power levels of 140 kW and 120 kW, the rate of temperature increase on the inner surface of the cladding is slower at a power level of 100 kW. The time for the copper alloy cladding to begin melting increases from 3.8 s to 7.8 s when comparing power levels of 140 kW and 100 kW. The time for the temperature at the metallurgical bond interface to reach 1400 °C increases from 7.5 s to 13.9 s.

The temperature distribution of the cylindrical workpiece at different cladding time points is shown in Figure 17 when simulating a power level of 80 kW. At 9 s of heating, the maximum temperature on the cylindrical workpiece can reach 963 °C. At 11.7 s of heating, the maximum temperature reaches 1095 °C, and the radial temperature distribution of the substrate and cladding exhibits similar color patterns. At 19.7 s of heating, the temperature at the metallurgical bond interface reaches around 1400 °C, and the radial temperature distribution of the substrate and cladding shows similar color patterns, approaching the melting point of the cylindrical steel substrate.

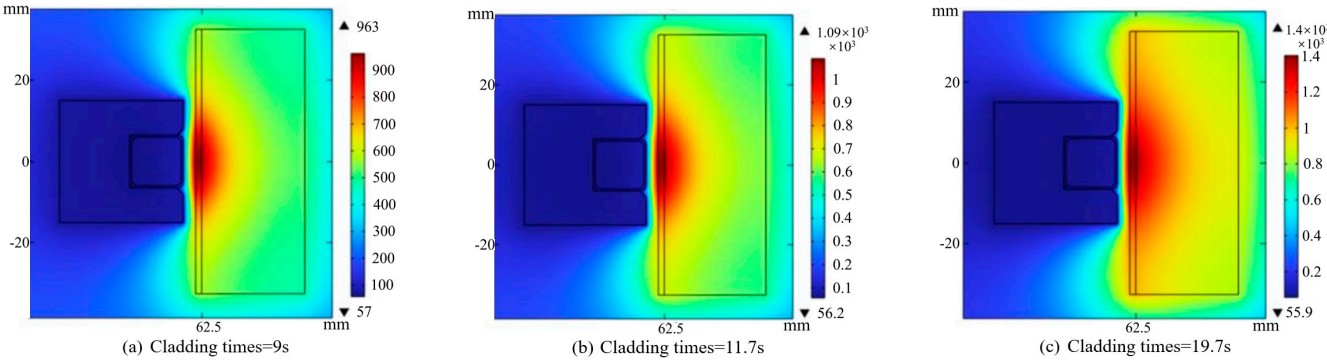

**Figure 17.** Temperature distribution at different melting times at a power of 80 kW.

The temperature cycle curves for various points on the cylindrical workpiece at a simulated current frequency of 120 kHz and a power level of 80 kW are shown in Figure S10. From the graph, it can be observed that the temperature rise rates at different parts of the cylindrical workpiece are all slow. At 11.7 s of induction heating, the cladding temperature just reaches the melting point. However, when the power level is increased to 120 kW, the melting temperature of the cladding can be reached in just 5.4 s. Therefore, it can be inferred that reducing the power level will prolong the heating time required for the workpiece to reach the desired temperature, thereby reducing the efficiency of inductively cladding copper alloy on the inner surface of the cylinder.

(4)    The influence of cladding thickness on the temperature field

To investigate the influence of cladding thickness on high-frequency induction heating, numerical simulations were conducted under the following conditions: power level of 120 kW, frequency of 120 kHz, heating time of 10 s, and a constant 3 mm gap between the workpiece and the cladding. Copper alloy claddings with thicknesses of 0.5 mm, 1.0 mm, 1.5 mm and 2.0 mm were set for the study. The transient temperature distributions of the cylindrical workpiece at 3 s of induction heating for different cladding thicknesses are shown in Figure S11. As shown in the figure, when the heating time is the same, the substrate temperature decreases with the increase of cladding thickness. The influence of cladding thickness on the induction melting process mainly includes: an increase in thickness is equal to the same amount of energy used to heat more substances, and it is obvious that the temperature rise is slower when the cladding is thicker at the same time. An increase in cladding thickness will result in a longer energy transfer time required to heat up to the melting temperature, causing thermal damage to the substrate. When the cladding thickness is small, there is less energy loss in all directions, and the temperature rises quickly.

The temperature distributions when reaching the melting point for different cladding thicknesses are shown in Figure 18. From Figure 18a–d, it can be observed that the time required to reach cladding melting is 2.2 s, 4.5 s, 6.7 s and 8.5 s for cladding thicknesses of 0.5 mm, 1.0 mm, 1.5 mm and 2.0 mm, respectively. This indicates that a thicker cladding requires more time to reach its melting temperature, resulting in a more significant thermal impact on the substrate region and a higher likelihood of thermal damage.

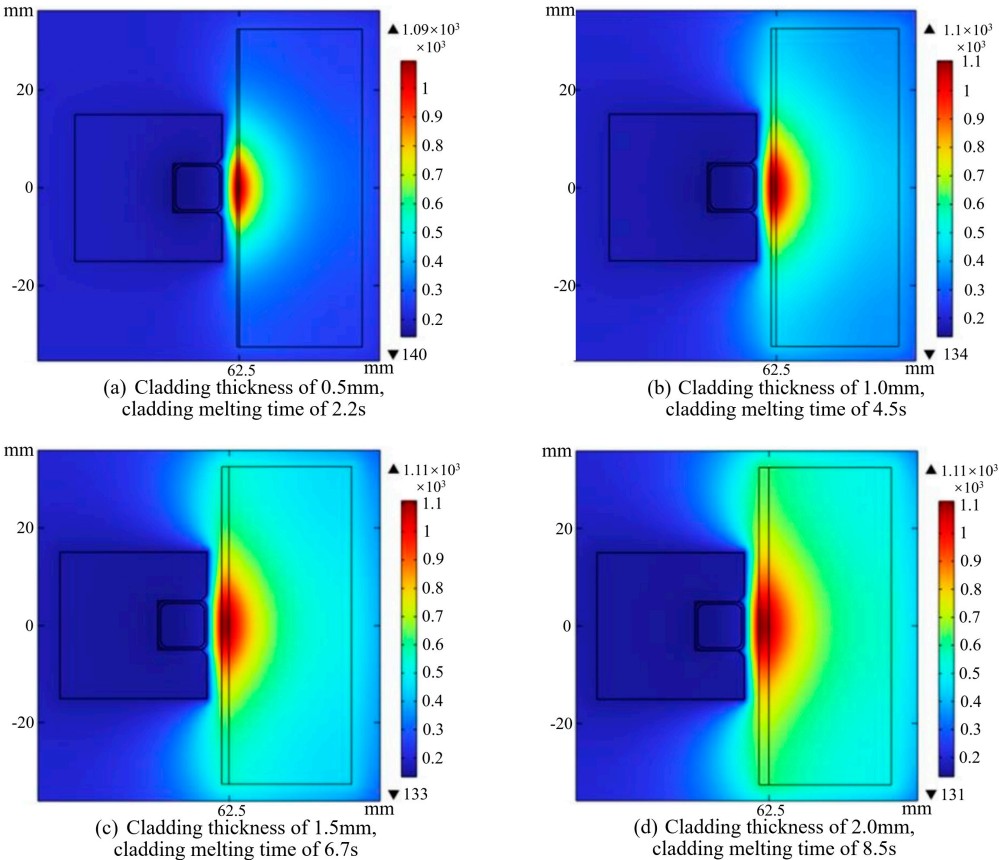

**Figure 18.** Temperature distribution map of claddings with different thicknesses reaching melting point.

The radial temperature cycle curve of the cylindrical workpiece when the temperature at the metallurgical bond interface reaches the melting point is shown in Figure 19. From the graph, it can be observed that the cladding with a thickness of 0.5 mm exhibits the smallest temperature difference within the cladding. Starting from the position of 100 mm, where the cladding enters the substrate region, the heat transfer from the cladding to the substrate occurs through conduction, and the substrate has poorer thermal conductivity compared to the cladding. As a result, the temperature reduction in the substrate is most significant when the cladding thickness is 0.5 mm, while the temperature reduction is less pronounced when the cladding thickness is 2.0 mm. Therefore, taking into account the required melting time for the comprehensive cladding and the thermal damage caused to the cylindrical substrate during the cladding process, the optimal cladding thickness of 1.5 mm is selected.

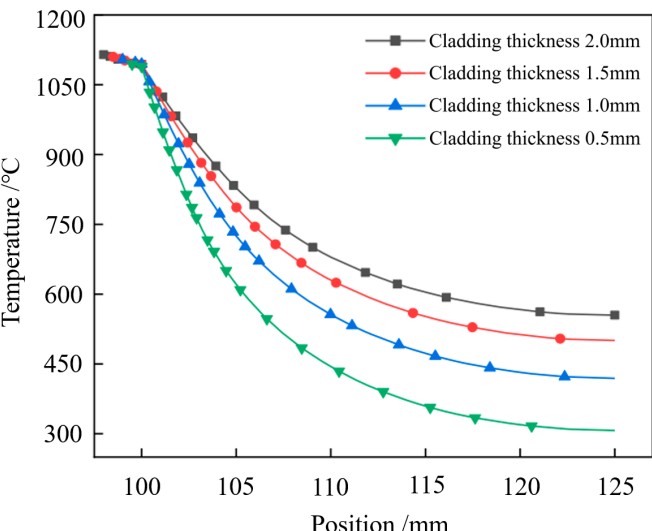

**Figure 19.** Radial thermal cycle curve of the workpiece when the metallurgical joint reaches melting point.

(5)    The influence of coil cross-sectional dimensions on the temperature field

In practical induction cladding heating processes, it is crucial to select coil cross-sectional dimensions that meet the requirements in order to ensure cladding quality. Since the coil cross-section is square, a study was conducted to investigate the impact of coil cross-sectional dimensions on HIC. Under the conditions of a power level of 120 kW, coil outer diameter of 116 mm, and frequency of 120 kHz, coil cross-sectional side lengths of 8 mm, 9 mm, 10 mm and 11 mm were set to simulate the effect of coil cross-sectional dimensions on induction heating performance. Figure 20 shows the temperature distribution maps after 2 s of heating for different coil cross-sectional dimensions. From the graph, it can be observed that the maximum temperature of the workpiece initially increases and then decreases with the increase in cross-sectional side length. When the cross-sectional side length increases from 8 mm to 10 mm, the maximum temperature of the workpiece increases from 689 °C to 715 °C. However, when the cross-sectional side length increases from 10 mm to 11 mm, the maximum temperature decreases from 715 °C to 690 °C.

In order to better understand the influence of different coil cross-sectional side lengths on induction heating effects, the temperature distribution map in Figure 20 was modified in Figure S12, where the regions with temperatures above 600 °C are displayed in red, while regions below 600 °C are left blank. From the graph, it can be observed that as the coil cross-sectional side length increases from 8 mm to 10 mm, the red area representing temperatures above 600 °C in the cladding expands. This suggests that the molten pool width during cladding melting is wider for a coil cross-sectional side length of 10 mm compared to 8 mm. However, when the cross-sectional side length increases from 10 mm to 11 mm, the red area in the cladding decreases, indicating a narrower molten pool width during cladding melting. This is due to the fact that with an increase in cross-sectional side length, the induction heating area expands, leading to enhanced heat dissipation and slower temperature rise.

Figure S13 shows the temperature distribution maps of the workpiece during induction heating at 6.5 s for different coil cross-sectional side lengths. The highest temperatures in Figure S13a–d are 1043 °C, 1089 °C, 1112 °C, and 1065 °C, respectively, with temperature differences in the cladding regions of 16 °C, 15 °C, 15 °C, and 16 °C. When the side length of the coil section increases from 8 mm to 10 mm, the highest temperature changes by 46 °C, and when it increases from 10 mm to 11 mm, the highest temperature changes by 45 °C. However, the temperature range of the copper alloy cladding remains nearly constant at around 16 °C, indicating that changing the side length of the coil section has a significant impact on the cylindrical steel substrate.

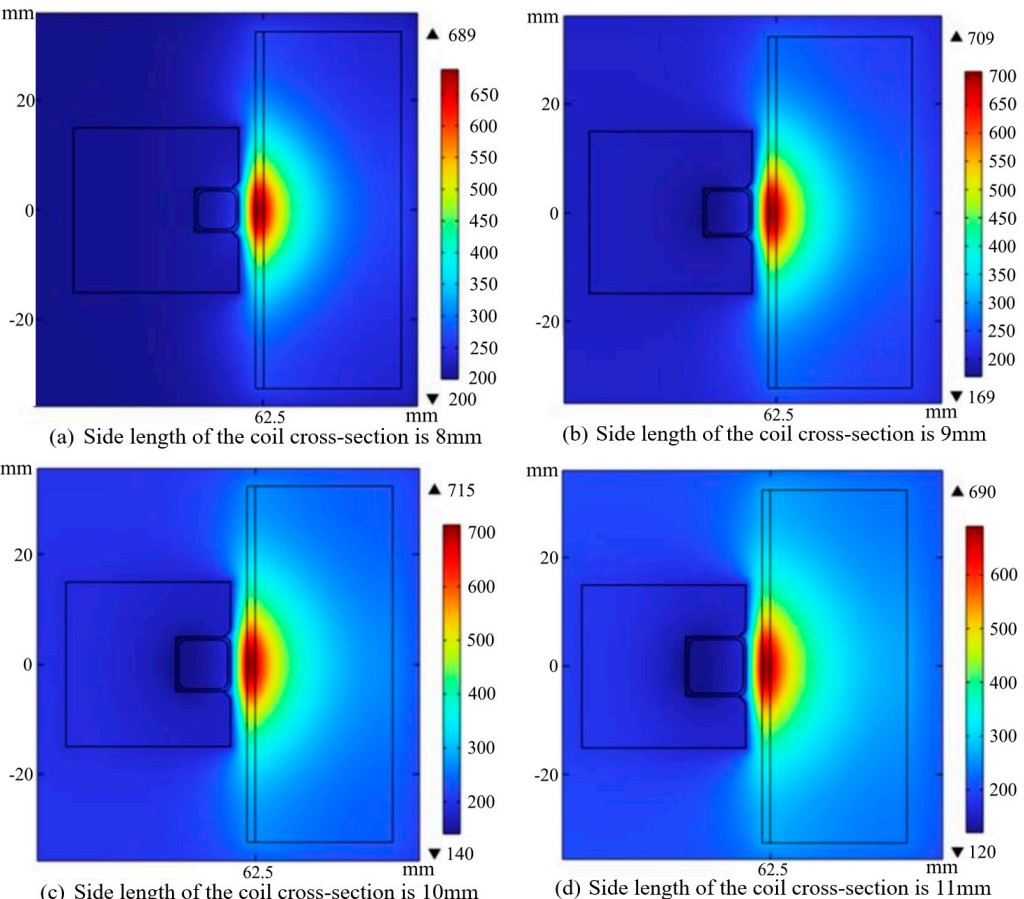

**Figure 20.** Temperature cloud map of the workpiece during 2 s fusion of different cross-sectional edge lengths.

Figure 21 shows the molten pool width of the copper alloy cladding when the metallurgical bond reaches the melting temperature under different coil cross-sectional side lengths during induction heating. From Figure 21a–d, it can be observed that the copper alloy cladding melts at 7.5 s, 7 s, 6.7 s and 7.2 s when the coil side lengths are 8 mm, 9 mm, 10 mm and 11 mm, respectively. The radial widths of the molten pools are 4.46 mm, 5.45 mm, 4.44 mm and 4.49 mm, respectively. It can be seen from the graph that the cladding reaches the melting point in the shortest time when the coil cross-sectional side length is 10 mm, and the widest radial molten pool width is achieved at a side length of 9 mm. Therefore, when selecting the optimal high-frequency induction coil cross-sectional side length, the melting time and molten pool width should be considered simultaneously.

The temperature cycle curves of the center point at the interface between the cladding and the substrate under induction cladding heating with different coil cross-sectional side lengths are shown in Figure 22. From the graph, it can be observed that the temperature at the center point increases with an increase in the coil cross-sectional side length. However, when the cross-sectional side length exceeds 10 mm, the rate of temperature increase actually decreases. This is because the heated area and the cooling area of the cylindrical workpiece increase with the increase in the coil cross-sectional side length, resulting in a slower heating rate of the workpiece.

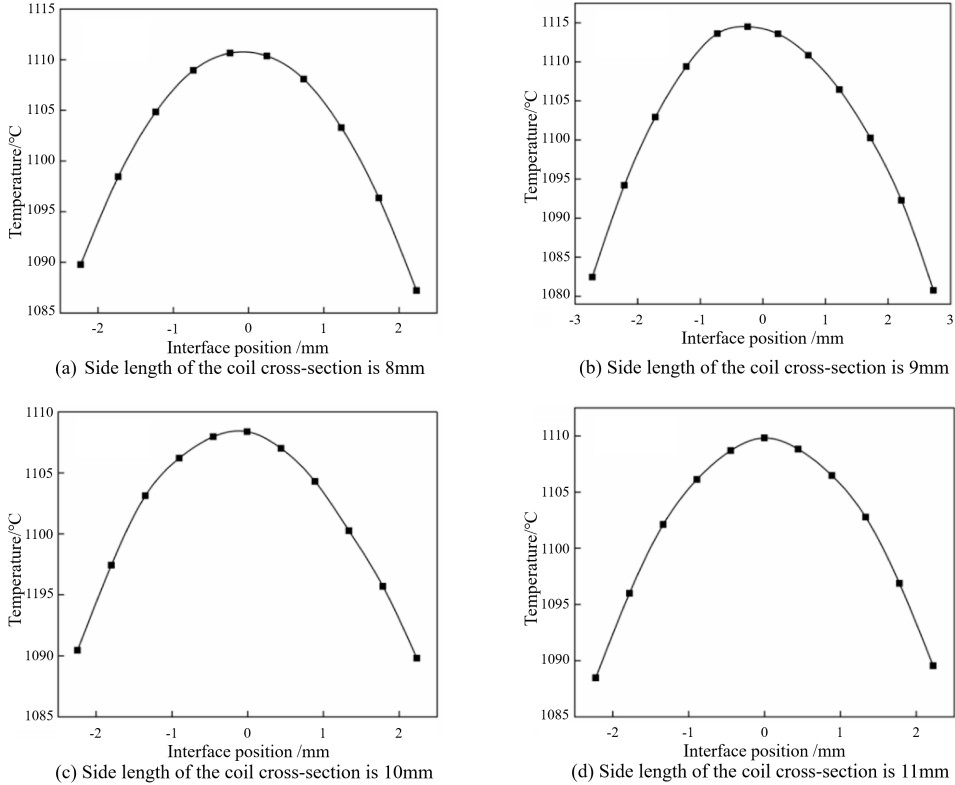

**Figure 21.** The width of the cladding melt pool when the metallurgical bonding surface reaches the melting point.

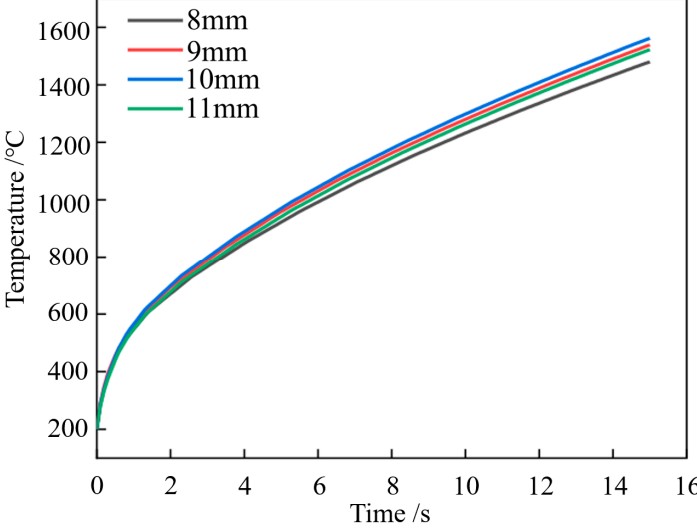

**Figure 22.** Temperature cycling curves of E under different cross-sectional lengths of coils.

## 3. Experimental Study on HIC of Copper Alloy on the Inner Wall of Cylindrical Workpieces

In this study, an HIC experimental platform was constructed to address the issue of "gravity flow" in inner wall claddings by utilizing the centrifugal force generated by the rotation of the workpiece. Optimal process parameters such as current frequency, coil gap, power supply, cladding thickness, and coil cross-sectional side length were selected for the induction cladding experiments. Furthermore, microstructural observations and hardness tests were conducted on the interface between the cladding and the substrate to confirm the completion of the induction cladding. Additionally, spectroscopic analysis, friction and

wear tests, and electrochemical corrosion tests were performed on the cladding to further verify the desired quality of the formed cladding.

### 3.1. Experimental System Composition

The experimental system for HIC on the inner wall of cylindrical workpieces primarily consists of a high-frequency induction heating power supply, a compressed air water cooling system, an infrared thermometer, a rotational and translational device for the cylindrical workpiece, and an induction heating coil with a magnetic guide. The specific equipment parameters are presented in Table S1. The water-cooling system includes a softened water circulation cooling system and an air-cooled box-type industrial chiller. The temperature detection system mainly comprises an infrared temperature sensor, laser temperature measurement, and a color screen backlight display, enabling color screen display, indication of the highest, lowest, and average temperatures, as well as long-distance temperature measurement and conversion and calculation of various temperature values. The rectifier bridge is used to convert the input three-phase AC power into DC voltage, providing overvoltage protection and smooth and stable current in the DC circuit. The inverter bridge converts the DC voltage into the required high-frequency AC voltage. By paralleling and resonating between the resonant capacitor and the transformer, it improves the power factor of the heated workpiece.

### 3.2. Experimental Process Design

An experimental study was conducted to deposit copper alloy claddings on the inner surface of a cylindrical workpiece using high-frequency induction melting, based on the results of numerical simulations. The experimental parameters are detailed in Table 4, and the chemical composition of the copper alloy cladding and the cylindrical steel substrate are presented in Table S2. During the cladding preparation, QJ102 solder paste was chosen as the melting aid, and a blend of polyvinyl alcohol (PVA) powder with a solubility of 88% and deionized water was used as the binder. The binder, solvent, and copper alloy powder were mixed to form a paste, which was evenly cold-applied onto the pre-treated inner surface of the cylindrical workpiece after being polished with different grades of sandpaper and cleaned with acetone. After drying the applied paste, the workpiece was placed in a drying oven and dried for 2 h at 200 °C, resulting in a cladding thickness of 1.5 mm.

**Table 4.** Experimental parameters.

| Experiment Number | Frequency/kHz | Power/kW | Gap/mm | Coil Cross-Section/mm |
|:---:|:---:|:---:|:---:|:---:|
| 1 | 120 | 100 | 2 | 9 |
| 2 | 120 | 120 | 3 | 10 |
| 3 | 120 | 140 | 4 | 11 |
| 4 | 150 | 100 | 3 | 11 |
| 5 | 150 | 120 | 2 | 9 |
| 6 | 150 | 140 | 4 | 10 |
| 7 | 200 | 100 | 3 | 10 |
| 8 | 200 | 120 | 2 | 11 |
| 9 | 200 | 140 | 4 | 9 |

### 3.3. Fitting of Experimental and Simulation Results

The temperature variation on the outer surface of the cylindrical substrate was measured using an infrared thermometer, as shown in Figure 23a, and the measured temperature trend matched the simulation results. Numerical simulation of the cylindrical workpiece was performed using high-frequency induction, resulting in a temperature cycle curve for point H on the outer surface of the workpiece, as depicted in Figure 23b. During the induction cladding experiment, the temperature of point H on the outer surface of the cylinder was measured using an infrared thermometer. The temperature data were processed using Origin2022 software, generating a scatter plot and a fitted curve for the

temperature at point H, as shown in Figure 23c. By combining the numerical simulation curve with the experimental temperature measurements, Figure 23d was obtained. It can be observed from the figure that the numerical simulation results closely matched the temperature measurements obtained from the induction cladding experiment, indicating the reliability of the numerical simulation. However, it should be noted that the presence of errors was attributed to factors such as the distance and ambient temperature affecting the temperature measurement of the outer surface of the cylinder using the infrared thermometer.

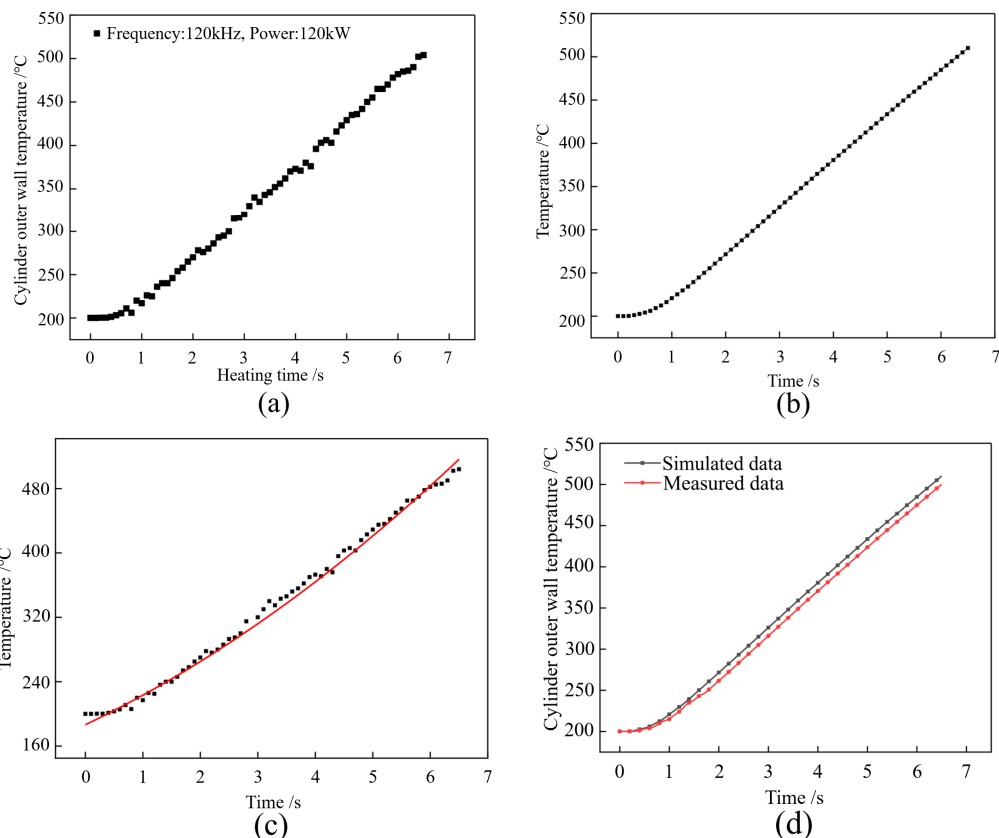

**Figure 23.** (**a**) Temperature curve of the outer surface of the substrate. (**b**) H-point temperature curve on the outer surface of the substrate. (**c**) H-point temperature scatter plot and its fitting curve. (**d**) Combination diagram of simulated curve and measured data.

### 3.4. Experimental Results

After the high-frequency induction-melted cladding samples were naturally cooled to room temperature, segmental line cutting was performed. As shown in Figure 24, the samples at various stages of treatment are displayed, and the cut surfaces were observed to evaluate the metallurgical bonding between the copper alloy cladding and the cylindrical steel substrate. Figure 25 illustrates the microstructure image of the cladding-substrate interface under the second set of conditions in Table 4. In Figure 25a, a good bond between the cladding and substrate is observed, with the copper alloy cladding exhibiting a stable compound structure. This promotes grain refinement, enhances fluidity, and reduces the formation of pores and cracks. Some of the slag generated on the surface of the copper alloy cladding due to electromagnetic stirring and thermal convection is processed during subsequent treatments. However, a few pores, impurities, and substrate overheating phenomena are observed in the claddings shown in Figure 25b–d.

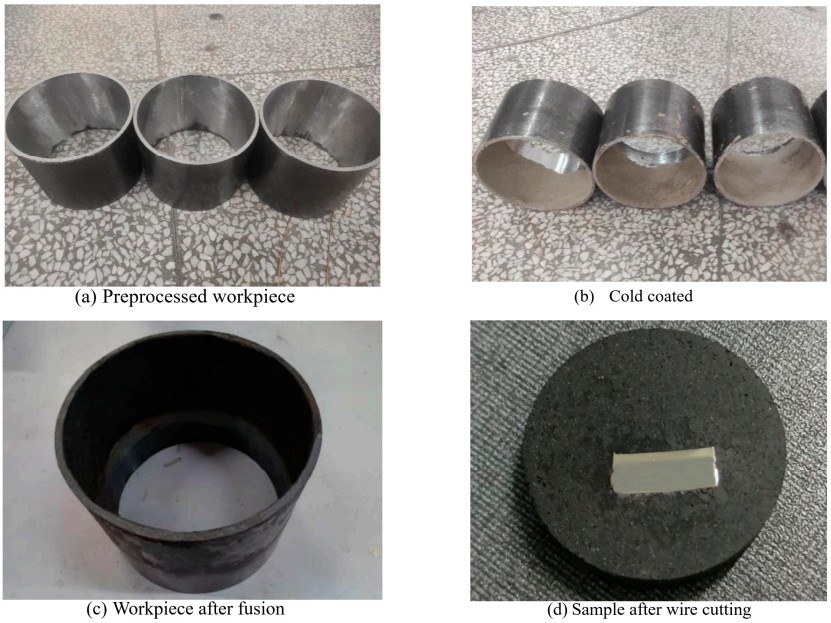

|  |  |
|---|---|
| (a) Preprocessed workpiece | (b) Cold coated |
| (c) Workpiece after fusion | (d) Sample after wire cutting |

**Figure 24.** Cylindrical workpiece processing process.

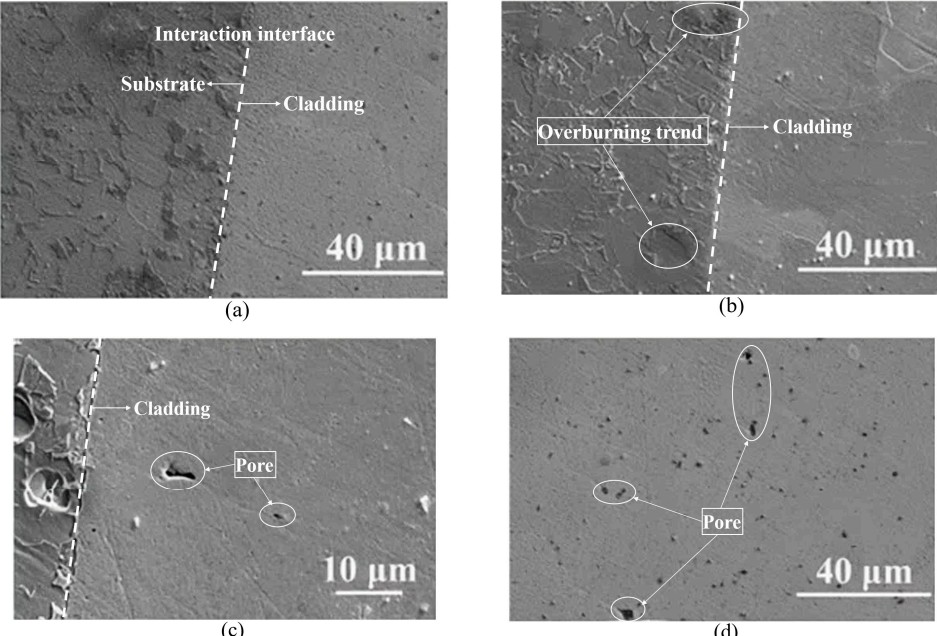

**Figure 25.** Microscopic structure diagram of metallurgical bonding surface. (**a**) Defect-free interactive interface; (**b**) Matrix overburning; (**c**) Microstructure observation of cladding layer; (**d**) Pore defect of cladding layer.

Microhardness measurements were performed on the obtained samples under different working conditions using an FM-700 digital microhardness tester. A load of 200 N was applied to each sample for a testing duration of 15 s. Along the radial direction of the copper alloy cladding surface, a test point was taken every 100 μm from the substrate, totaling 20 test points. The microhardness value for each point was calculated using the median of five tests. The results of the microhardness tests are shown in Figure S14, and the microhardness values at the bond interface are listed in Table S3. The analysis of microhardness in the inductive cladding sample is shown in Figure 26. From Figure 26, it can be observed that the microhardness of the copper alloy cladding layer, both with and without defects, is significantly higher than the microhardness of the 27SiMn steel

substrate. However, the copper alloy cladding layer with defects (Sample 2, Sample 3) exhibits larger fluctuations in hardness distribution and poorer uniformity, with a relatively larger heat affected zone (HAZ) in the substrate. In the defect-free copper alloy cladding layer (Sample 1), the hardness near the interface is higher than that of the cladding layer and the cylindrical workpiece substrate. This is because the temperature in the bond region reaches the melting point of the copper alloy, forming a quenched layer on the surface of the cylindrical workpiece substrate. This improves the fluidity of the cladding layer melt pool, promoting the diffusion of elements between the cladding and the substrate.

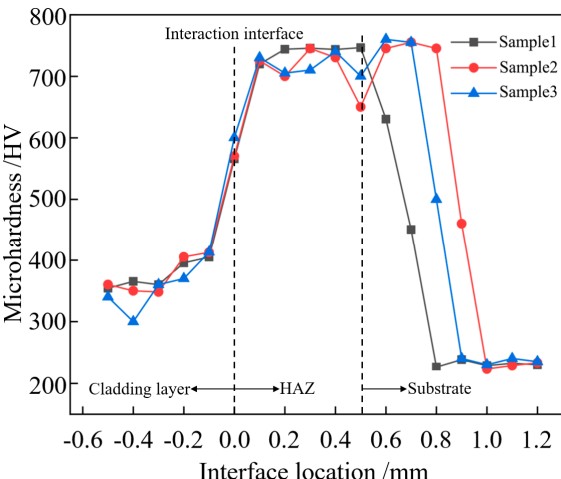

**Figure 26.** Microhardness analysis of induction cladding samples.

The MFT-R4000 high-speed reciprocating wear and friction tester was used to conduct friction and wear tests on three samples and a cylindrical workpiece substrate. The diameter 6 mm $Si_3N_4$ balls were used as the counter-specimens for both the claddings and the substrate. The friction test was performed under the conditions of a 5 N applied load, a reciprocation frequency of 2 Hz, a wear length of 10 mm, and a wear time of 20 min. Prior to the friction and wear tests, the sample surfaces were pre-ground and polished. Figure 27 shows the average friction coefficient curves of different samples, in which the copper alloy cladding layer without defects shows the most stable friction coefficient, and the minimum average friction coefficient is 0.0205. However, the friction coefficient of other samples fluctuates significantly with the increase of wear time, and is higher than that of copper alloy cladding without defects.

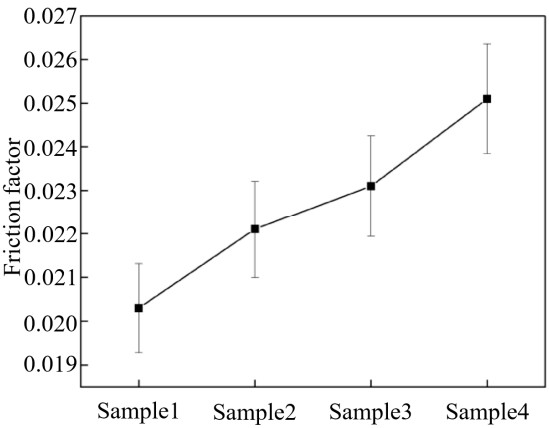

**Figure 27.** Average friction coefficient of different claddings.

The surface topography and wear volume of the cladding were analyzed using the MicroXAM-800 non-contact optical profilometer. Figure 28 illustrates the surface morphol-

ogy of defect-free and defective claddings after 20 min of wear under a 5 N load. From Figure 28a, it can be observed that the wear surface of the defect-free cladding is smooth and even, with only a few shallow grooves present. In Figure 28b, the worn surface of the cladding appears relatively smooth, with shallow grooves, some minor delamination, and a significant amount of debris, indicating abrasive wear. Figure 28c shows that the worn surface of the cladding is relatively rough, exhibiting slight delamination, peeling, and plastic deformation, which are typical features of plastic deformation wear. Figure 28d illustrates the uneven wear on the surface of the defective cladding, characterized by deep grooves, layering, and extensive peeling, indicating severe abrasive and adhesive wear mechanisms. Among them, the width and depth of the wear track in Figure 28a are the smallest. This is because during the friction and wear test, heat accumulation leads to the formation of a hardened layer or oxide layer on the worn surface of the cladding, effectively reducing the contact between the mating surfaces and the cladding. This inhibits the propagation of microcracks in the mechanical mixed layer, thereby reducing the wear rate of the cladding and improving its wear resistance.

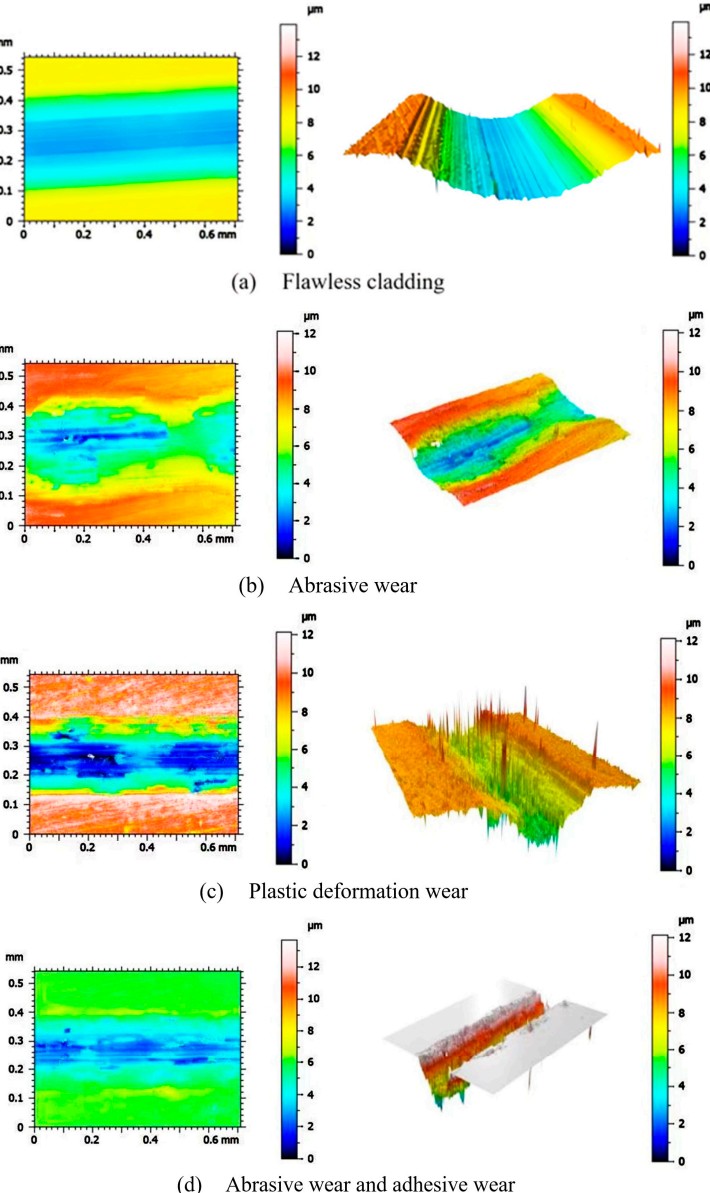

(a)  Flawless cladding

(b)  Abrasive wear

(c)  Plastic deformation wear

(d)  Abrasive wear and adhesive wear

**Figure 28.** Microscopic and three-dimensional morphology of friction and wear tests on different specimens.

To investigate the influence of copper alloy overlay claddings on the corrosion resistance of the substrate surface, the samples with defective overlay cladding, defect-free overlay cladding, and cylindrical steel substrate were immersed in a 3.5 wt% NaCl solution. Figure 29 shows the polarization curves of the samples, and Table 5 provides the analysis results of the electrode corrosion potential and corrosion current density for each sample based on the polarization curves.

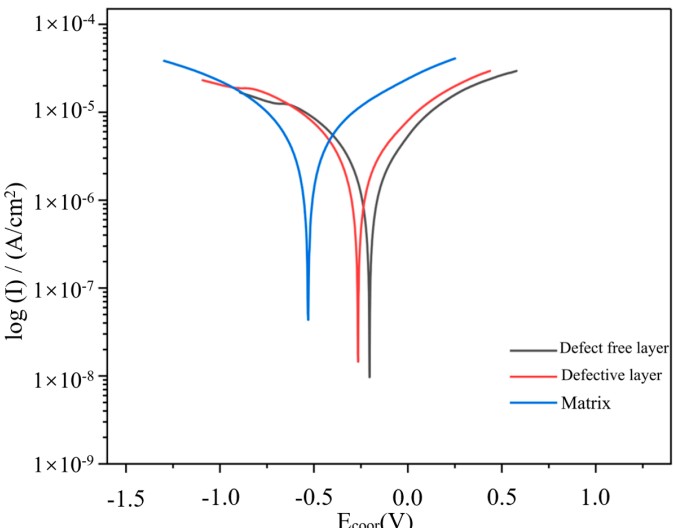

**Figure 29.** Tafel polarization curves of matrix and different samples in 3.5 wt% NaCl solution.

**Table 5.** Electrochemical corrosion test data.

| Type | Matrix | Defective Cladding Layer | Defect-Free Cladding Layer |
|---|---|---|---|
| $E_{corr}$ (V) | −0.532 | −0.265 | −0.224 |
| $I_{corr}$ ($\mu A/cm^2$) | 8.461 | 2.781 | 0.446 |

In the electrochemical corrosion test, the corrosion rate of the claddings is determined by the corrosion current density, which can be reflected by the current density in the polarization curve. From Figure 29 and Table 5, it can be observed that there are significant differences in the corrosion resistance among the samples in the 3.5 wt% NaCl corrosion solution. The corrosion potential of the cylindrical steel substrate is −0.532 V. When the copper alloy overlay cladding has defects, the corrosion potential decreases to −0.265 V. Meanwhile, the defect-free copper alloy overlay cladding has a corrosion potential of only −0.224 V. The corrosion potential on the surface of the overlay cladding shows a noticeable forward shift compared to the substrate. The corrosion current density of the cylindrical substrate is 8.461 $\mu A/cm^2$, while the corrosion current densities of the defective and defect-free overlay claddings are 2.781 $\mu A/cm^2$ and 0.446 $\mu A/cm^2$, respectively. Both claddings exhibit improved corrosion resistance compared to the cylindrical steel substrate.

## 4. Conclusions

In order to enhance the comprehensive performance of the inner wall of cylindrical workpieces and overcome the limitations of other surface modification techniques, this study employs numerical simulation analysis of HIC. An experimental platform for HIC of copper alloy on the inner wall of cylindrical workpieces has been established. The preparation method for the cold cladding and the optimized combination of process parameters during HIC have been determined. The focus of this study was on investigating the bond strength between the substrate and the cladding layer. The following conclusions were drawn:

(1) During the HIC process, the current density of the high-frequency induction coil gradually decreases from the inner side to the outer side due to the skin effect. However, by introducing a magnetic conductor, the current density on the outer side of the induction coil increases, thereby meeting the requirements for induction cladding on the IWC workpieces.

(2) Based on numerical simulation and analysis of experimental data, an optimized set of process parameters for HIC of copper alloy on the inner wall of a 125 mm diameter 27SiMn cylindrical workpiece has been determined. The optimal parameter combination consists of a power supply of 120 kW, a heating frequency of 120 kHz, a cladding thickness of 1.5 mm, a 3 mm gap between the cylindrical workpiece and the induction coil, and an outer diameter of 116 mm for the heating coil.

(3) The cladding preparation method has been established through experimental investigations. It entails the blending of copper alloy powder with QJ102 solvent, along with polyvinyl alcohol acting as a binder, and deionized water. The resulting mixture is stirred to achieve a paste-like consistency for cold cladding. Subsequently, the prepared paste is applied onto the substrate surface and allowed to dry. To address the challenge of gravity flow and promote the formation of the cladding layer, the cylindrical workpiece is subjected to rotation during the induction cladding process, utilizing centrifugal force.

(4) Built upon the optimized process parameters for HIC of copper alloy on the IWC workpiece, an HIC experiment was performed utilizing the parameter combination derived from numerical simulations. Physical examinations demonstrated that under these parameter conditions, the cladding layer exhibited reduced cracking, a refined grain structure, and a uniform distribution of hard phases. In addition, the average hardness of the cladding layer reaches 400 HV. Compared with the substrate, the average friction coefficient of the defect-free cladding layer is 0.0205, the corrosion potential is $-0.224$ V, and the corrosion current density is 0.446 $\mu A/cm^2$. The cladding layer shows enhanced wear resistance and corrosion resistance.

(5) The manual pre-placement of powder introduces significant random errors to the cladding process, making it challenging to reproduce the exact shape and powder compaction. Therefore, the future development of in situ powder feeding and automated cladding techniques, along with extensive research on various auxiliary technologies, holds crucial importance in achieving precise control over the cladding process.

**Supplementary Materials:** The following supporting information can be downloaded at: https://www.mdpi.com/article/10.3390/coatings14040458/s1, Figure S1: The physical parameters of the 27SiMn steel matrix change with temperature trends, Figure S2: The physical parameters of copper alloys change with temperature trends, Figure S3: Temperature distribution at different frequencies at 3 s, Figure S4: Temperature distribution at different frequencies at 9.5 s, Figure S5: Temperature distribution under different outer diameters of induction coils at 2 s, Figure S6: Temperature distribution under different outer diameters of induction coils at 4 s, Figure S7: Simulate the thermal cycle curve of each point on a cylindrical workpiece, Figure S8: Simulate the thermal cycle curve of each point on the 2-cylindrical workpiece, Figure S9: Simulate the thermal cycle curve of each point on the 3-cylindrical workpiece, Figure S10: Simulate the thermal cycle curve of each point on the 4-cylindrical workpiece, Figure S11: Temperature distribution of different cladding thicknesses at 3 s, Figure S12: Temperature cloud map of the workpiece during 2 s fusion of different cross-sectional edge lengths, Figure S13: Temperature cloud map of the workpiece during induction melting for 6.5 s, Figure S14: Schematic diagram of microhardness testing; Table S1: Experimental equipment parameters, Table S2: Chemical composition of copper alloy cladding and cylindrical steel substrate, Table S3: The microhardness value of the bonding interface.

**Author Contributions:** Conceptualization, L.H. and X.Z.; Data curation, R.P. and T.X.; Writing—original draft, Y.W. (Yafei Wang); Writing—review & editing, L.H., Z.Z., J.C. and Y.W. (Yue Wu); Visualization, J.G. All authors have read and agreed to the published version of the manuscript.

**Funding:** This research supported by the Qin Chuangyuan "Scientists+Engineers" Team Construction Foundation 2023KXJ-225, China National Natural Science Foundation 52204174, 52074210 and 52301103, China Postdoctoral Science Foundation 2022MD723828, Shaanxi Postdoctoral Science Foundation 2023BSHTBZZ44, Shaanxi University Youth Innovation Team Foundation 23JP096.

**Institutional Review Board Statement:** Not applicable.

**Informed Consent Statement:** Not applicable.

**Data Availability Statement:** Data are contained within the article and Supplementary Materials.

**Conflicts of Interest:** The authors declare no conflicts of interest.

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
