# Peer review of "A Fast Method of High-Frequency Induction Cladding Copper Alloy on Inner-Wall of Cylinder Based on Simulation and Experimental Study"

_coatings, doi:10.3390/coatings14040458_

Round 1
Reviewer 1 Report
Comments and Suggestions for Authors
1 The manuscript is too big. The modeling part is described in great detail.
2 Figure 27. It is not clear where the steel is, where the copper is. Discontinuities smaller than 1 µm are not pores. To reveal the microstructure, it is necessary to subject the samples to etching. Then examine the fusion line in detail and study the chemical composition of the transition layer. What will be a scientific research, and not a solution of a technical problem.
3 Figure 29. For the convenience of readers, “Interface location” should indicate not only mm, but also draw vertical lines corresponding to: the transition line from the base metal of the steel to the HAZ of the steel; from the steel HAZ to the copper fusion line; from the steel-copper fusion line to the copper HAZ; from the copper HAZ to the copper base metal. How this is done, for example, in work Fiber laser welding of austenitic steel and commercially pure copper butt joint.
4 Figure 29. I doubt that the microhardness of the copper base metal can be more than 200 HV, this should be double-checked.
Comments on the Quality of English Language-
Author Response
Dear Editor-in-Chief and reviewer 1,
Thank you for allowing a resubmission of our manuscript again, with an opportunity to address the reviewer 1’ comments. All the comments are very helpful in guiding our research and improving the manuscript and we have improved the original methodology, detailed the case study, and rewritten most part of the manuscript. The point-by-point response to reviewers is provided in this submission package as a separate file.
We are uploading (a) our point-by-point response to the comments (below) (response to reviewers), (b) an updated manuscript with red highlighting indicating content changes, blue highlighting indicating unchanging, and (c) a clean updated manuscript without highlights (PDF main document).
We confirm that this manuscript has not been published elsewhere and is not under consideration by another journal. All the authors have approved the manuscript and agreed with this submission. We have no conflicts of interest to disclose.
Thank you for your consideration.
Best regards,
Longlong He

Reviewer 2 Report
Comments and Suggestions for Authors
I have no significant comments on the submitted article "A Fast Method of high-frequency induction cladding Copper Alloy on inner-wall of cylinder based on Simulation and Experimental Study". I recommend only small comments to the authors to improve the quality of the outputs and results of the publication:
1. For such a long article, it is necessary to increase the number of cited literature
2. I recommend enlarging all pictures and graphs, respectively the description on the axes of the graphs, for better readability for everyone who will read this article
3. Complete the chapter "Discussion", in which you will link all the obtained results and compare them with the results of other authors.
4. The article is unnecessarily long, I recommend removing unnecessary information and focusing only on the results obtained. (remove photos of measuring devices) (useless table 6 - results are in figure 29.
Fig 30 chaotic useless graph
The presented article is unnecessarily long and somewhat chaotic (because there is too much information in it (33 pictures is really too much)). It is necessary to shorten it, please follow my instructions
Author Response
Dear Editor-in-Chief and reviewer 2,
Thank you for allowing a resubmission of our manuscript again, with an opportunity to address the reviewer 2’ comments. All the comments are very helpful in guiding our research and improving the manuscript and we have improved the original methodology, detailed the case study, and rewritten most part of the manuscript. The point-by-point response to reviewers is provided in this submission package as a separate file.
We are uploading (a) our point-by-point response to the comments (below) (response to reviewers), (b) an updated manuscript with red highlighting indicating content changes, blue highlighting indicating unchanging, and (c) a clean updated manuscript without highlights (PDF main document).
We confirm that this manuscript has not been published elsewhere and is not under consideration by another journal. All the authors have approved the manuscript and agreed with this submission. We have no conflicts of interest to disclose.
Thank you for your consideration.
Best regards,
Longlong He

Reviewer 3 Report
Comments and Suggestions for Authors
The manuscript entitled “A Fast Method of high-frequency induction cladding Copper Alloy on inner-wall of cylinder based on Simulation and Experimental Study” mainly focuses on the method of high-frequency induction cladding copper alloy on inner-wall of cylinder.
This manuscript is well written, with a formulated key problem of the corresponding research object. Nevertheless, I have a few comments on several aspects.
It is unclear (method, equipment) how the thermal trends of the physical parameters for the cylindrical steel substrate, shown in Figs. 3 and 4, were estimated?
Why the heating temperature for the modeling procedure during the induction cladding process was set between 1000°C and 1100°C? What is the melting point of the alloy used? Why was the temperature range from 1000 oC to 1100 oC chosen? Could this temperature range be narrower? What is the chemical composition of the corresponding copper alloy? Is it crystalline? Without these data, mathematical models are meaningless because they will be identical for all systems.
This manuscript uses supplementary material that was not made available for review.
I have not been able to relate the theoretical modelling to the experimental data. I also failed to capture reduced porosity and cracking in the photos of the samples. In the International Union of Pure and Applied Chemistry (IUPAC) classification of pore size, the micropore width is taken to not exceed about 2 nm (20 Ǻ), the mesopore width to be in the range 2–50 nm and the macropore width to be above about 50 nm. What kind of porosity did the authors of this paper want to capture?
Author Response
Dear Editor-in-Chief and reviewer 3,
Thank you for allowing a resubmission of our manuscript again, with an opportunity to address the reviewer 3’ comments. All the comments are very helpful in guiding our research and improving the manuscript and we have improved the original methodology, detailed the case study, and rewritten most part of the manuscript. The point-by-point response to reviewers is provided in this submission package as a separate file.
We are uploading (a) our point-by-point response to the comments (below) (response to reviewers), (b) an updated manuscript with red highlighting indicating content changes, blue highlighting indicating unchanging, and (c) a clean updated manuscript without highlights (PDF main document).
We confirm that this manuscript has not been published elsewhere and is not under consideration by another journal. All the authors have approved the manuscript and agreed with this submission. We have no conflicts of interest to disclose.
Thank you for your consideration.
Best regards,
Longlong He

Round 2
Reviewer 3 Report
Comments and Suggestions for Authors
No comments.